# Nutrient restriction synergizes with retinoic acid to induce mammalian meiotic initiation in vitro

Xiaoyu Zhang[1], Sumedha Gunewardena[1] & Ning Wang [1✉]

The molecular machinery and chromosome structures carrying out meiosis are frequently conserved from yeast to mammals. However, signals initiating meiosis appear divergent: while nutrient restriction induces meiosis in the yeast system, retinoic acid (RA) and its target Stra8 have been shown to be necessary but not sufficient to induce meiotic initiation in mammalian germ cells. Here, we use primary culture of mouse undifferentiated spermatogonia without the support of gonadal somatic cells to show that nutrient restriction in combination with RA is sufficient to induce *Stra8*- and *Spo11*-dependent meiotic gene and chromosome programs that recapitulate the transcriptomic and cytologic features of in vivo meiosis. We demonstrate that neither nutrient restriction nor RA alone exerts these effects. Moreover, we identify a distinctive network of 11 nutrient restriction-upregulated transcription factor genes, which are associated with early meiosis in vivo and whose expression does not require RA. Our study proposes a conserved model, in which nutrient restriction induces meiotic initiation by upregulating key transcription factor genes for the meiotic gene program and provides an in vitro platform for meiotic induction that could facilitate research and haploid gamete production.

[1] Department of Molecular and Integrative Physiology, University of Kansas Medical Center, Kansas City, KS, USA. ✉email: nwang2@kumc.edu

Sexual reproduction depends on the formation of haploid gametes through meiosis, which exchanges replicated parental chromosomes through homologous recombination during meiotic prophase, followed by two rounds of successive divisions to generate haploid gametes carrying novel genetic constitution[1]. To initiate meiosis, germ cells must activate a specialized meiotic gene program that implements the intricate chromosomal events during meiosis prophase[2]. From yeasts to mammals, meiosis-specific chromosome structures exhibit remarkable evolution conservation, including synaptonemal complex assembly for chromosome synapsis and homologous recombination through formation and subsequent repair of meiotic DNA double-strand breaks (DSBs). Moreover, many genes underlying these events are often evolutionarily related or—conserved (reviewed in ref. [1]). For instances, Spo11 encodes a DNA topoisomerase-like enzyme that catalyzes meiotic DSB formation[3,4]. Dmc1 encodes a meiotic recombinase that repairs DSBs by searching for allelic DNA sequences on the homologous chromatids[5,6]. Hormad genes encode meiosis-specific chromosome factors (Hop1 in yeasts and Hormad1/2 in mammals) that are critical for synapsis and DSB formation and repair[7,8].

Despite these evolutionary conservations in meiotic genes and structures, the overarching signal to initiate meiosis appears divergent. In yeasts, the transition from mitotic to meiotic cell cycles is induced by a metabolic cue, nutrient restriction. Subsequently, nutrient-sensing pathway induces the expression of inducer of meiosis 1 (IME1), which encodes a single master transcriptional activator for meiotic genes, including Spo11, Dmc1, and Hop1 (ref. [9,10]). In mammalian germ cells, meiotic initiation requires retinoic acid (RA), the most biologically active metabolite of vitamin A. RA is a morphogen essential for growth and development in chordate animals[11]. In female oogenesis, RA is synthesized primarily in the mesonephric ducts to which the embryonic ovaries are attached[12]. In male spermatogenesis, RA is produced by both meiotic and somatic cells in testes[13]. RA induces meiotic gene program primarily by activating stimulated by retinoic acid gene 8 (Stra8) (ref. [14,15]). However, it is puzzling that, although the RA signaling, including STRA8 by itself or with its cofactor MEIOSIN, is necessary for meiotic initiation[16,17], neither RA nor STRA8 is sufficient, suggesting that additional signal(s) is required to work with RA to induce meiotic initiation in mammals. This represents a significant gap of knowledge in understanding the molecular mechanism underlying this fundamental biological process essential for to sexual reproduction.

Moreover, although long-term male germline stem cell (GSC) culture has been successfully developed in several mammalian species[18–20], a direct means to induce meiotic initiation in these cells in vitro remains as an unmet challenge[21]. As an example, this was be achieved by reintroducing cultured male GSCs back into the seminiferous tubules on the organ culture system[22], suggesting that gonadal somatic cell support is indispensable for this process. This prevents faithful recapitulation of meiosis and reconstitution of spermatogenesis in vitro under a defined culture condition, which possesses enormous value in investigating the delicate process of meiosis as well as in vitro production of haploid gametes that assist animal and, ultimately, human reproduction[23].

Here we report that nutrient restriction, inducer of yeast meiosis, in combination with RA is sufficient to induce meiotic initiation in primary culture of mouse spermatogonia or SSC culture that faithfully recapitulates the transcriptomic and cytologic features of in vivo meiotic initiation. Neither nutrient restriction nor RA treatment alone possesses this effect. Moreover, we show that nutrient restriction upregulates a set of transcription factor (TF) genes, whose expressions are not regulated by RA and are associated with early meiosis in vivo. In addition to establishing a culture system for robust, efficient, and faithful meiotic induction, our study provides mechanistic insights into meiotic initiation in mammalian germ cells by suggesting a conserved role of nutrient-sensing pathway in this process.

## Results

**Nutrient restriction in combination of RA induces meiotic initiation**. Our recent work in Stra8-deficient mice reveals that an autophagy-inducing factor is engaged on meiosis-initiating germ cells[24]. Interestingly, nutrient restriction, the aforementioned inducer of yeast meiosis, is perhaps the most potent inducer of autophagy[25]. Thus, we asked whether nutrient restriction might have a conserved role in inducing meiotic initiation in mammalian germ cells. To test this hypothesis, we established primary culture of undifferentiated spermatogonia (referred to as spermatogonial stem cell or SSC culture) by using neonatal mouse testicular cells in C57BL/6 X DBA/2 F1 hybrid background (Supplementary Fig. 1a; see below for transcriptomic analyses). The composition of the complete SSC medium is shown in Supplementary Table 1.

Earle's Balanced Salt Solution (EBSS), Hank's Balanced Salt Solution (HBSS), and phosphate buffered saline (PBS) are commonly used buffers to starve culture cells. Thus, we tested applying nutrient restriction to SSC culture by adding 90% EBSS, HBSS, and PBS to the complete SSC medium. Since cells became detached from culture dish in the medium containing HBSS or PBS, but tolerated well in the medium containing EBSS, we used EBSS to induce nutrient restriction in all following studies (referred to as "nutrient restriction medium"; Supplementary Table 1). A direct cellular response to nutrient restriction is autophagy[25]. We show that this nutrient restriction medium induced autophagy activation in SSC culture, suggesting that it is sufficient to apply nutrient restriction to cells (Supplementary Fig. 1b).

Co-treatment of nutrient restriction and RA (referred to as "NRRA") for 2 days triggered a distinct morphology of cell size enlargement in SSC culture, suggesting cellular differentiation (Fig. 1a, Supplementary Fig. 1c). Importantly, NRRA induced a significant activation of essential meiotic genes, including Spo11, Dmc1, and Sycp3 (Fig. 1b). Sycp3 encodes a lateral element of synaptonemal complex[26]. Consistently, phosphorylated histone H2AX (γH2AX) shows that DNA DSBs were most profoundly formed in cells from NRRA-treated culture (Fig. 1c, Supplementary Fig. 1d). These effects were not observed in SSC cultures treated with RA or NR alone (Fig. 1a–c). Similar effects of NRRA in inducing Spo11, Dmc1, and Sycp3 expression were also observed in F9 premeiotic cells (Supplementary Fig. 2). Moreover, the meiotic origin of these DSBs was confirmed by DMC1 staining (Fig. 1d). In addition, cells from SSC culture generated from Spo11-deficient mice do not exhibit DMC1 foci upon NRRA treatment, constituting genetic evidence that NRRA-induced meiotic DSB formation in vitro requires Spo11 (Fig. 1d). Cultured cells exhibited rapid loss of PLZF expression, a marker for undifferentiated spermatogonia[27,28], under the condition of RA and NRRA, which is consistent with the role of RA in inducing spermatogonial differentiation (Supplementary Fig. 1e). Nutrient restriction alone did not downregulate PLZF, suggesting a specific role for NR in synergizing with RA to induce meiotic initiation.

Transcriptomic analysis shows that NRRA treatment for 2 days induced a gene expression pattern distinct from the treatment of RA or NR alone in SSC culture (Fig. 1e, Supplementary Fig. 3, Supplementary Data 1). Four clusters of gene sets were identified by unsupervised hierarchical clustering (UHC) (Supplementary

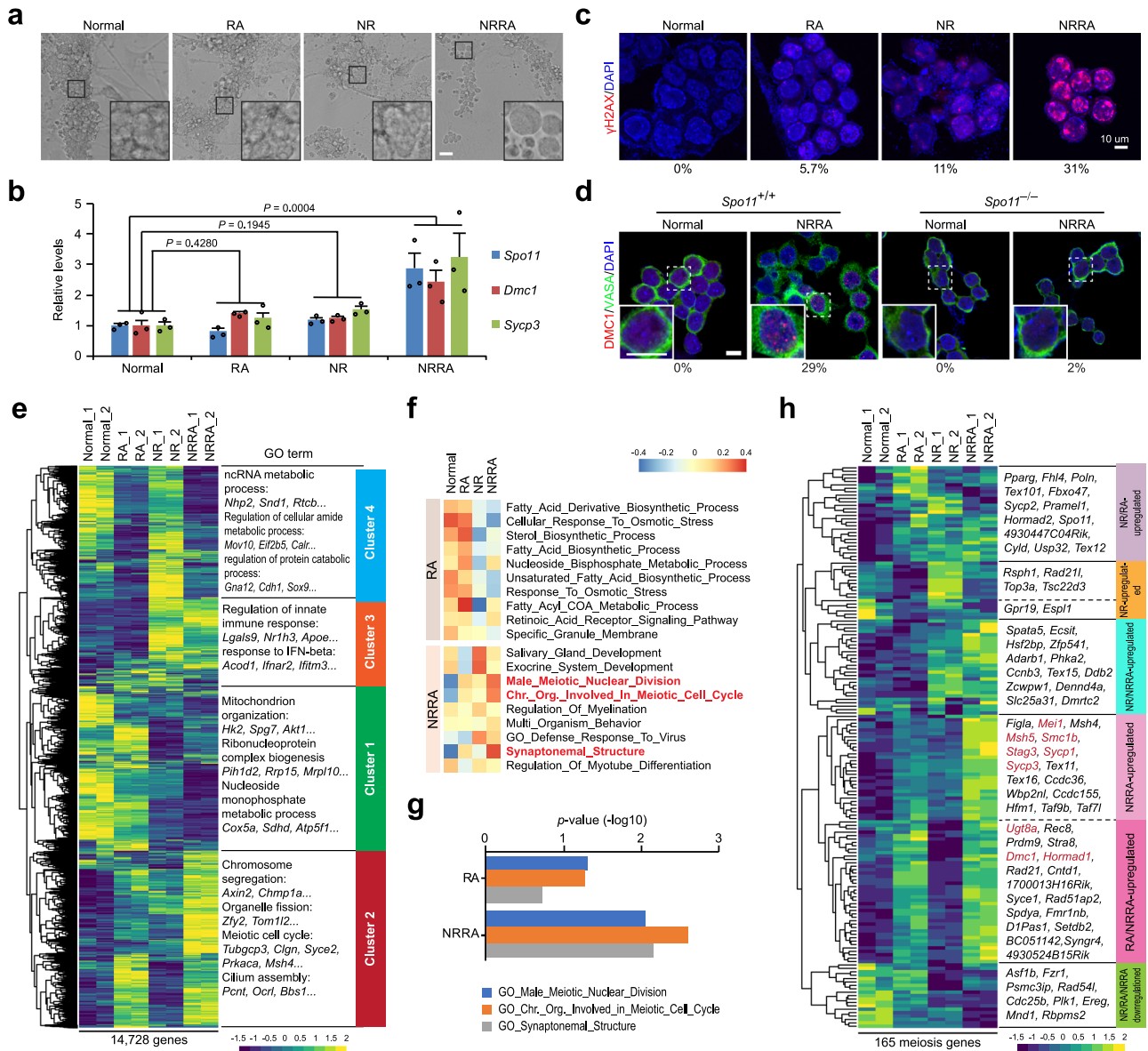

**Fig. 1 Nutrient restriction synergizes with RA to induce meiotic initiation in SSC culture. a** Bright-field images of SSC culture with indicated treatments for 2 days. Scale bars, 50 μm. **b** Relative expression of Spo11, Dmc1, and Sycp3 against Gapdh analyzed by qRT-PCR in SSC culture with indicated treatments for 2 days. Results shown represent mean ± SEM. *$P < 0.05$. $n = 3$ independent cultures. n.s. no significant. **c** Immunostaining for γH2AX (red) and DAPI (blue) with indicated treatment for 2 days. Scale bars, 10 μm. **d** Immunostaining for DMC1 (red), DDX4 (green), and DAPI (blue) in Spo11$^{+/+}$ and Spo11$^{-/-}$ SSC culture. Scale bars, 10 μm. In **c**, **d**, the numbers below the immunofluorescent images indicate the percentage of cells seen with the relevant staining under different conditions. **e** (Left) UHC and heatmap of gene expression in SSC cultures with indicated treatments for 2 days. (Right) Top GO enrichments with representative genes in each cluster. **f** GSVA analysis for indicated treatments. In the heatmap, rows are defined by the selected gene sets, and columns by consensus scores for each treatment. Group enriched gene sets are highlighted by different color. **g** Bar plot shows $p$ value of GO functions between RA and NRRA treatment. **h** UHC and heatmap for the expression of 165 early meiosis-associated genes. STRA8-dependent genes characterized in a previous study in ref. [2] are shown in red. $P$ value by (**b**) two-way ANOVA with Turkey's multiple comparison test and (**g**) GO functions. Source data are provided in Source Data file. Data are representative of **c**, **d** three independent experiments.

Data 2). Notably, genes in Cluster 2, which appear to be genes upregulated by RA and NRRA, are enriched with genes bearing gene ontology (GO) functional terms related to meiosis (Fig. 1e, Supplementary Fig. 4). In contrast, Cluster 1, 3, or 4 is enriched with genes for "mitochondrion organization" (Cluster 1, mostly nutrient restriction-downregulated genes), "regulation of innate immune response" (Cluster 3, mostly nutrient restriction-upregulated genes), "ncRNA metabolic process" (Cluster 4, mostly RA-downregulated genes) (Supplementary Fig. 4). Consistently, nonparametric and unsupervised gene set variation analysis

(GSVA) and hallmark gene set analysis revealed stronger positive correlations of meiosis-related pathways (male meiotic nuclear division, chromosome organization in meiotic cell cycle, and synaptonemal structure) and gametogenesis-related (spermatogenesis) in genes upregulated by NRRA than those upregulated by RA (Fig. 1f, g, Supplementary Fig. 5) (Supplementary Data 3). These data together suggest that nutrient restriction in combination with RA induces meiotic gene program and *Spo11*-dependent meiotic DSB formation in vitro. Importantly, this culture condition does not require gonadal somatic cell support.

To examine whether the effect of NRRA in inducing meiotic gene program in other genetic background, we established SSC culture using testes from neonatal CD1 inbred mice. RNA-seq show that cultured cells in CD1 background exhibited comparable transcriptomic changes including activation of essential meiotic genes, such as *Dmc1* and *Sycp3*, in response to NRRA treatment (Supplementary Fig. 6). These data suggest that the effect of NRRA in inducing meiotic gene program does not depend on a specific genetic background.

To delineate the role of nutrient restriction in activating meiotic gene program, we assembled a set of 193 meiosis genes by combining two collections of genes that were previously found to be associated with mammalian early meiosis during mouse[2] and human[29] spermatogenesis (Supplementary Fig. 7). In addition, we also included *Gm4969*, which encodes MEIOSIN, a recently identified transcriptional cofactor for STRA8 required for meiotic prophase program[17]. UHC divided 165 differentially expressed genes (DEGs) among these 194 genes into five major clusters (Fig. 1h, Supplementary Data 4). Nutrient restriction appears to play four roles on the expression of key meiotic genes by: (1) inducing a subset of meiotic gene expression that was not regulated by RA, such as *Rad21l*; (2) initiating the activation of a subset of meiotic genes, which was further enhanced by RA, such as *Zfp541*; (3) synergizing with RA to induce the activation of a subset of meiosis gene expressions, such as *Sycp1*, *Sycp3*, *Mei1*, *Msh5*, and *Stag3*; and (4) augmenting the expression of a subset of meiotic genes induced by RA, such as *Dmc1*, *Smc1b*, *Ugt8a*, *Stag3*, *Hormad1*, and *Gm4969*. Notably, many meiotic genes whose expressions are STRA8-dependent in vivo require nutrient restriction for activation in vitro (Fig. 1h) (ref. [2]). Together, these data suggest crucial roles of nutrient restriction by itself or by synergizing with RA in inducing meiotic gene program; in the absence of nutrient restriction, RA alone is not sufficient (see the effect of in vivo RA treatment on meiotic gene expression in neonatal testes in Supplementary Fig. 8) (ref. [30]).

**scRNA-sequencing analysis of NRRA-induced meiotic initiation and progression in vitro.** To examine NRRA-induced meiotic initiation at the single-cell resolution and its potential to support meiotic progression, we first treated SSC culture with NRRA for 2 days to induce meiotic initiation, followed by a medium that is supposed to promote meiotic progression (referred to as "meiotic progression" medium; Supplementary Table 1) for an additional 1 day (a total of 3-day treatment) or 2 days (a total of 4-day treatment) (Fig. 2a). This "meiotic progression" medium is formulated by omitting GDNF and FGF2, cytokines that maintain stem cell renewal, from the complete SSC medium to allow for spermatogonial differentiation. To support meiotic progression, we included melatonin ($10^{-7}$ M), follicle-stimulating hormone (FSH) (200 ng/ml), transforming growth factor (TGF)-β (10 ng/ml), bovine pituitary extract (BPE) (50 ng/ml), and dihydrotestosterone (DHT) ($10^{-6}$ M). SSC culture treated with complete SSC medium was used as a control (Fig. 2a). Single-cell RNA-sequencing (scRNA-seq) was then performed using the 10× Genomics platform.

A total of 25,607 cells were sequenced from these samples. Using stringent quality control, 18,088 cells were selected for further analysis (Supplementary Fig. 9). All of the cells were pooled to perform clustering analysis, which revealed four major cell clusters based on their distinct gene expression patterns (Fig. 2b). These four clusters were subsequently annotated by using known marker genes (Fig. 2c, Supplementary Fig. 10). Cluster 0 appears to be undifferentiated spermatogonia (e.g., *Gfra1*, *Etv5*). Clusters 1, 2, and 3 appear to be differentiating spermatogonia/pre- and meiotic spermatocytes at progressively

advanced meiotic stages with upregulated expression of spermatogonial differentiation (e.g., *Ccnb1*, *Cyp26a1*, *Sohlh1*) and meiotic genes (e.g., *Sycp1*, *Prdm9*). Cluster 4 appears to be mostly somatic feeder cells (mouse embryonic fibroblast or MEF) due to the presence of fibroblast marker gene expression (*s100a4*) and absence of germ cell marker gene (*Ddx4*) expression. Progressive upregulation of meiotic genes and downregulation of undifferentiated spermatogonia genes on each time points were confirmed by qRT-PCR analysis (Supplementary Fig. 11).

Analysis of the distribution of the different cell clusters at each time point demonstrate a dramatic transition of cell populations based on their transcriptomic features (Fig. 2d). Analysis of GO functional terms for DEGs revealed rapid changes in their cellular processes (Fig. 2e, Supplementary Data 5). Cluster 0 is enriched in genes involved in regulation of cellular amide metabolic process (e.g., *Sox4*), stem cell division (e.g., *Zbtb16*, *Etv5*), etc (Supplementary Fig. 12a). Cluster 1 is enriched in genes involved in the regulation of gene silencing (e.g., *Hist1h1e*), mitochondrial electron transport, NADH-ubiquinone (e.g., *Park7*), retinoic acid and metabolic process/response (e.g., *Stra8*), etc (Supplementary Fig. 12b). Cluster 2 is enriched in genes involved in reproduction in multicellular organisms (e.g., *Sycp3*), sperm–egg recognition (e.g., *Ly6k*), etc (Supplementary Fig. 12c). Cluster 3 is enriched in genes for meiotic cell cycle (e.g., *Prdm9*), cytoplasmic translation (e.g., *Rpl18a*), translation initiation (e.g., *Eif1*), chromosome condensation (e.g., *Dnajc19*), etc (Supplementary Fig. 12d).

Importantly, cells from Cluster 1 to Cluster 3 exhibit a progressive and coordinated upregulation of meiotic genes bearing mouse meiosis-related GO terms, including DNA double-strand break formation, histone variants, meiotic nuclear division, synapsis (Supplementary Fig. 13). Notably, meiotic genes that are fully (*Dmc1*, *Hormad1*, *Mei1*, and *M1ap*) or partially (*Smc1b*, *Stag3*, *Sycp1*, *Sycp2*, *Sycp3*, *Ugt8a*, and *Meioc*) (Fig. 2f, Supplementary Fig. 14) dependent on STRA8 expression displayed progressive upregulation from Cluster 1 to Cluster 3. Since activation of RA signaling with *Stra8* expression is not sufficient to induce their expression in vitro and in vivo (Fig. 1e, Supplementary Fig. 8) (ref. [30]), these data suggest that nutrient restriction is a critical signal to synergize with RA in activating meiotic genes.

To assess the relationship between NRRA-induced meiotic initiation and progression in vitro with those during in vivo meiosis, we used a scRNA-seq dataset of mouse spermatogenesis (Supplementary Fig. 15) (ref. [31]). Principal component analysis (PCA) analysis shows that the transcriptional profiles of Clusters 0–3 associate with the stages of meiotic initiation (leptotene, the first stage of meiotic prophase) and progression (zygotene and early pachytene) during in vivo spermatogenesis (Fig. 2g). Subsequently, hclust and differential gene correlation analysis demonstrate independently that cell population in Cluster 0 from in vitro meiosis correlates with undifferentiated spermatogonia during in vivo spermatogenesis, cell population in Cluster 1 with differentiating spermatogonia/preleptotene spermatocytes, cell population in Cluster 2 with leptotene/zygotene spermatocytes, and cell population in Cluster 3 with early pachytene spermatocytes (Fig. 2h, i). A hallmark event that occurs in male germ cells during pachytene stage is meiotic sex chromosome inactivation (ref. [32]). Similarly, we observed a progressive decline in sex chromosome gene transcription from Cluster 0 to Cluster 3, supporting that cells in Cluster 3 have reached to a pachytene-like stage at the transcriptomic level (Fig. 2j).

Moreover, early meiosis during in vivo spermatogenesis follows a stepwise pattern without lineage branching (Fig. 2k, Supplementary Fig. 15) (ref. [31]). To examine the transcriptional dynamics of in vitro meiotic initiation and progression, we performed pseudo-time

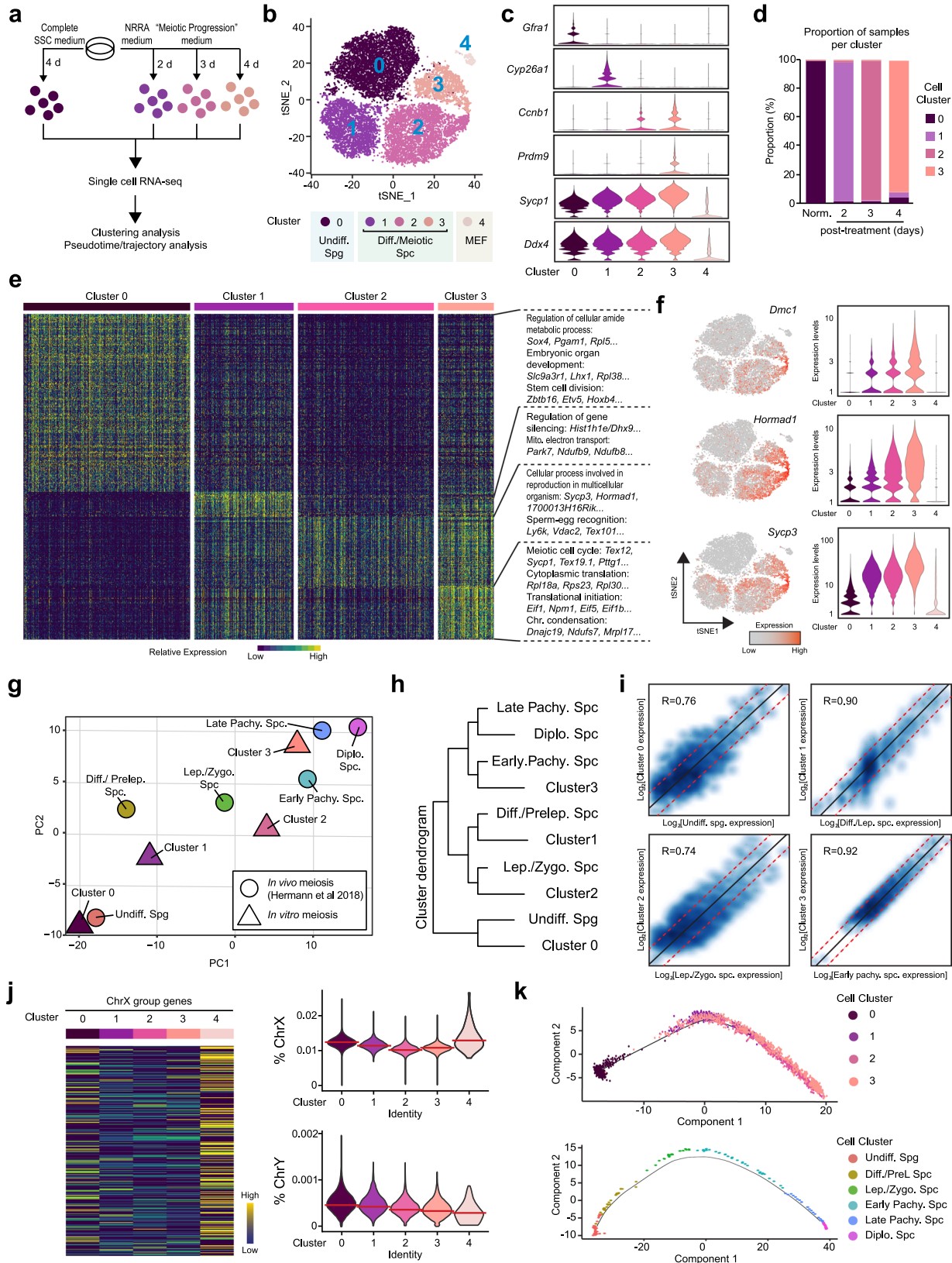

analysis, which shows that the projected timeline recapitulated early meiosis during in vivo spermatogenesis (Fig. 2k, Supplementary Fig. 16). The pseudo-time indicates that Cluster 0 is mainly at the start of the projected timeline trajectory, that Clusters 1, followed by Cluster 2, is positioned in the middle, and that Cluster 3 is at the end. Consistently, expression of genes specific to

undifferentiated spermatogonia was located preferentially at the beginning of the trajectory, while expression of genes functionally involved in many essential meiotic programs (e.g., cohesion, DNA DSB formation, chromosome segregation) was located preferentially towards to the end of the trajectory, which further supports the validity of the analysis (Supplementary Figs. 17, 18).

**Fig. 2 scRNA-seq analysis of SSC culture during NRRA-induced meiotic initiation and progression. a** Workflow of scRNA-seq experiment. Number of cells collected: 6884 (0 day), 6936 (2 day), 6200 (3 day), and 5587 (4 day). **b** A t-distributed Stochastic Neighbor Embedding (tSNE) plot for analyzed cells. Cluster 0–3 were germ cells and Cluster 4 was somatic cells. Number of cells selected for analysis: Cluster 0 (6487 cells), Cluster 1 (5286 cells), Cluster 2 (3860 cells), Cluster 3 (2170 cells), and Cluster 4 (285 cells). **c** Violin plots showing the expression level of representative genes in each cluster. **d** A bar plot showing the proportion of the different cell clusters at different time points. **e** A heatmap showing the expression of marker genes and GO functions in each cluster. **f** Gene expression patterns of indicated genes on tSNE plots and violin plots. Expression levels are calculated by UMI counts. **g** PCA analysis showing the relationship between single-cell clusters from NRRA-treated culture in vitro ($n = 4$ clusters) and early mouse spermatogenesis in vivo ($n = 6$ clusters) based on the transcriptional profiles of commonly expressed genes ($n = 7803$ genes). **h** Hierarchical clustering showing the relationship between single-cell clusters from NRRA-treated culture in vitro and early mouse spermatogenesis in vivo. "NR/RA-upregulated" indicates genes that were upregulated by NR or RA. "NRRA-upregulated" indicates that were upregulated by NRRA (not upregulated by NR or RA alone). **i** Scatter plots comparing marker genes expression profile between in vitro clusters and in vivo clusters. **j** (Left) Heatmap of sex chromosome genes in Clusters 0–3. (Right) Violin plots showing percentage of X and Y chromosome genes profiles in Clusters 0–3. The median is shown by a red line. **k** Scatter plots showing cells along the projected pseudo-time in Cluster 0–3 (upper) and in indicated clusters from early mouse spermatogenesis in vivo (lower).

**NRRA induces chromosome synapsis**. Chromosomal synapsis and recombination are key events of meiosis prophase I. To examine whether NRRA-induced activation of meiotic gene program supports synaptonemal complex formation, we conducted co-localization staining for cultured cells in dish and their chromosome spreads using SYCP3, a lateral element of synaptonemal complex[26], and SYCP1, a late component of central element of synaptonemal complex[33] (Fig. 3a–c). While nuclear accumulation of SYCP3 appeared between day 1 and 2 (preleptotene-like stage) when cultured cells were treated with NRRA medium, intermittent SYCP3 staining began to appear in chromosomal axis from day 3 when cells were switched to the "meiotic progression" medium, suggesting a leptotene-like stage. Between day 3 and 4, SYCP1 foci were detected, suggesting an early zygotene-like stage. Between day 4 and 5, gradual co-localization of SYCP3 and SYCP1 appeared, suggesting a late zygotene-like stage. Between day 5 and 6, extensive co-localization of SYCP3 and SYCP1 were observed, suggesting that cultured cells had reached to an early pachytene-like stage. The kinetics of in vitro meiosis appears to be consistent with in vivo meiosis: first wave of meiosis begins from postnatal day 7 to 8 and pachytene spermatocytes are observed in testes at around postnatal day 15. Together, these suggest that NRRA-induced meiotic initiation supports meiotic progression.

To further confirm that nutrient restriction and RA are both required for synaptonemal complex formation, we treated SSC culture with vehicle, RA, nutrient restriction medium, and NRRA medium for 2 days, followed by parallel treatment of "meiotic progression" medium for 4 days (Fig. 3d, e). We found that only cultured cells treated with NRRA reached to early pachytene stage. In contrast, SSC culture that was treated with the "meiotic progression" medium did not exhibit any meiotic signs. Most cultured cells receiving RA treatment alone were arrested at a preleptotene-like stage, in that they did not exhibit SYCP1 staining but only showed nuclear SYCP3 staining without SYCP3 loading onto chromosome axis. These data suggest that the "meiotic progression" medium, which lacks GDNF and FGF2, does not have any effect on meiotic initiation. Moreover, SSC culture treated with nutrient restriction medium alone before being treated with "meiotic progression" medium displayed no SYCP3 nor SYCP1 staining, confirming that nutrient restriction by itself does not induce meiosis. Together, these data are in agreement with our transcriptomic analysis that nutrient restriction and RA are both required for proper activation of meiotic gene program (Fig. 1e–h).

**NRRA-induced meiotic gene program requires Stra8**. Stra8 is a best characterized gatekeeper of meiotic initiation in vertebrates[14]. To examine whether NRRA-induced meiotic initiation meets this genetic requirement, we generated primary

spermatogonia culture using Stra8-deficient mice. Consistent with the essential role of STRA8 in meiotic DSB formation in vivo[14,34] (Supplementary Fig. 19), Stra8-deficient culture did not exhibit DMC1 foci upon NRRA treatment, suggesting that NRRA-induced in vitro meiotic DSB formation requires Stra8 (Fig. 4a). Moreover, RNA-seq analysis shows that WT, Stra8-deficient, and Spo11-deficient cultures exhibited similar transcriptome profiles in normal medium (Fig. 4b). However, upon parallel NRRA treatment to induce meiotic initiation for 2 days, UHC analysis identified that Stra8-deficient culture, but not Spo11-deficient culture, failed to upregulate a cluster of genes, which is enriched for meiosis-related GO terms, including meiotic cell cycle, meiosis I, synapsis (Fig. 4b, Supplementary Data 6). GSVA analysis shows that STRA8 sits at the foundation of NRRA-induced meiotic gene program in vitro, in that gametogenesis- and meiosis-related pathways were not activated in Stra8-deficient culture upon NRRA treatment (Fig. 4c). This is consistent with the essential role of STRA8 in activating gene program of meiotic prophase[2,14,34], and confirms that lack of meiotic DSBs in Stra8-deficient and Spo11-deficient cultures resulted from discrete mechanisms. In Spo11-deficient culture, despite normal activation of meiotic gene program, meiotic DSBs were not formed due to absence of SPO11 (Fig. 1d), the enzyme directly responsible for this process[4,35]. In Stra8-deficient culture, genes with GO term "retinoid acid receptor signaling pathway" was activated, suggesting that the failure Stra8-deficient culture to activate meiotic gene program did not result from a lack of RA signaling response (Fig. 4c). Moreover, STRA8-dependent meiotic genes (Sycp3, Mei1, Dmc1, Stag3, Sycp2) were only upregulated in WT, but not Stra8-deficient, culture (Fig. 4d, Supplementary Fig. 20, Supplementary Data 7). And the genes only upregulated in Stra8-deficient culture bears no meiosis-related GO terms, which is consistent with the robust meiotic initiation arrest phenotype of Stra8-deficient germ cells in vivo[14,34]. Additionally, our system reveals that many genes associated with undifferentiated spermatogonia were not downregulated in Stra8-deficient culture (e.g., Pou5f1, Etv5), which is in line with the role for STRA8 in promoting spermatogonial differentiation[13] (Fig. 4e, Supplementary Fig. 21, Supplementary Data 7).

**NRRA-induced chromosome synapsis requires Stra8 and Spo11**. To examine the genetic requirement for Stra8 and Spo11 in synaptonemal complex formation in vitro, we subjected Stra8-deficient and Spo11-deficient cultures to NRRA treatment for 2 days, followed by the treatment of "meiotic progression" medium for 4 days to induce synapsis formation. Stra8-deficient culture exhibited nuclear SYCP3 staining without showing SYCP3 loading onto meiotic chromosomes (Fig. 5a, b). SYCP1 was not detected in Stra8-deficient germ cells (Fig. 5a, b). These data suggest that Stra8 is required for meiotic progression

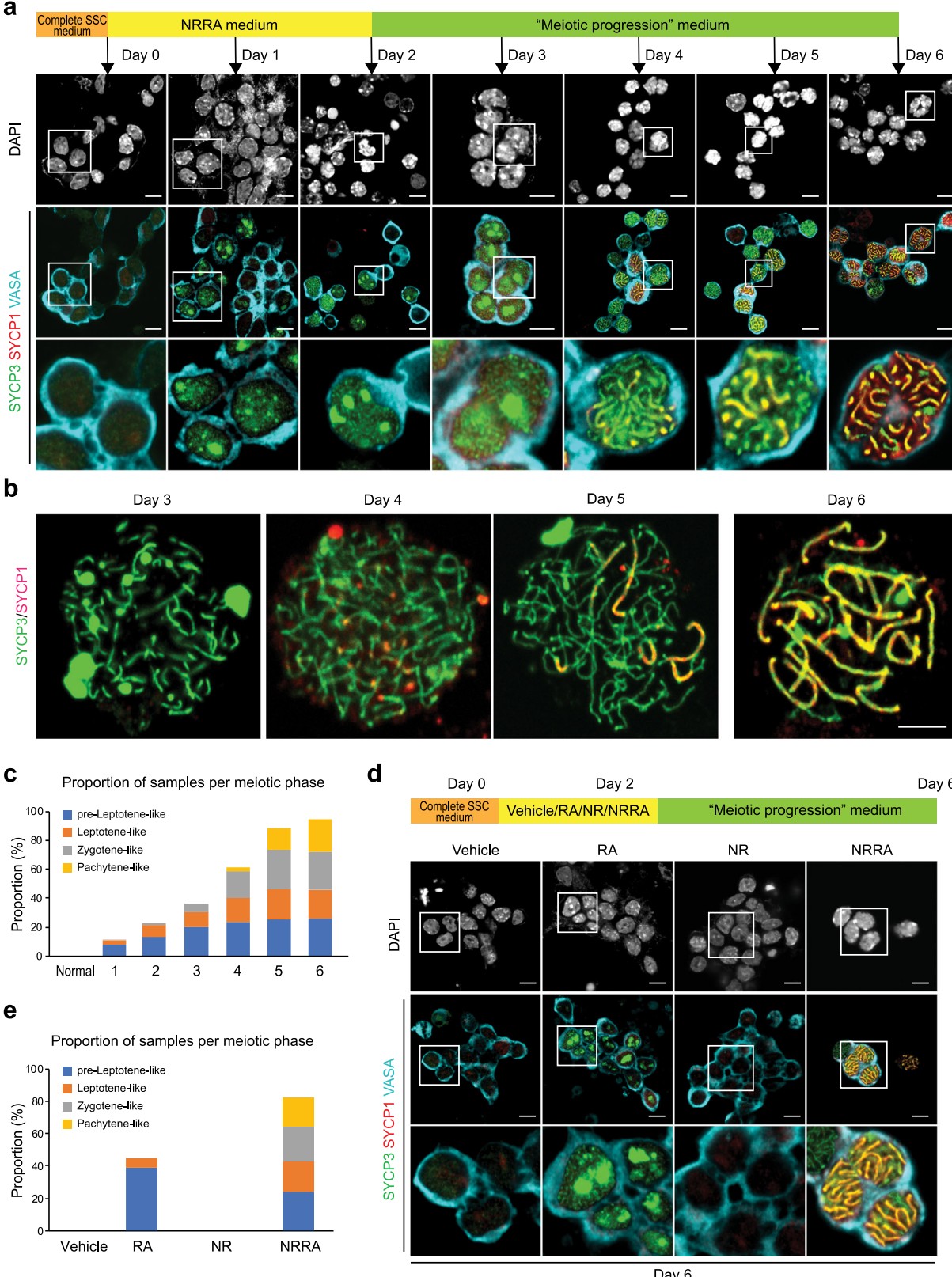

**Fig. 3 NRRA-induced meiotic initiation supports meiotic progression. a** DAPI staining and immunostaining for SYCP1 (red), SYCP3 (green), and VASA (cyan) in SSC cultures with indicated treatment. Scale bars, 10 μm. **b** Representative chromosome spreads stained by SYCP1 (red) and SYCP3 (green) from SSC cultures following indicated treatment are shown. **c** Percentages of germ cells at the indicated stages in culture on each day following indicated treatment (*n* = 200 cells). Scale bar, 5 μm. **d** DAPI staining and immunostaining for SYCP1 (red), SYCP3 (green), and VASA (cyan) in SSC cultures with indicated treatment for 2 days, followed by treatment of meiotic progression medium for 4 days. Scale bars, 10 μm. **e** Percentages of germ cells at the indicated stages in culture on each day following indicated treatment (*n* = 200 cells).

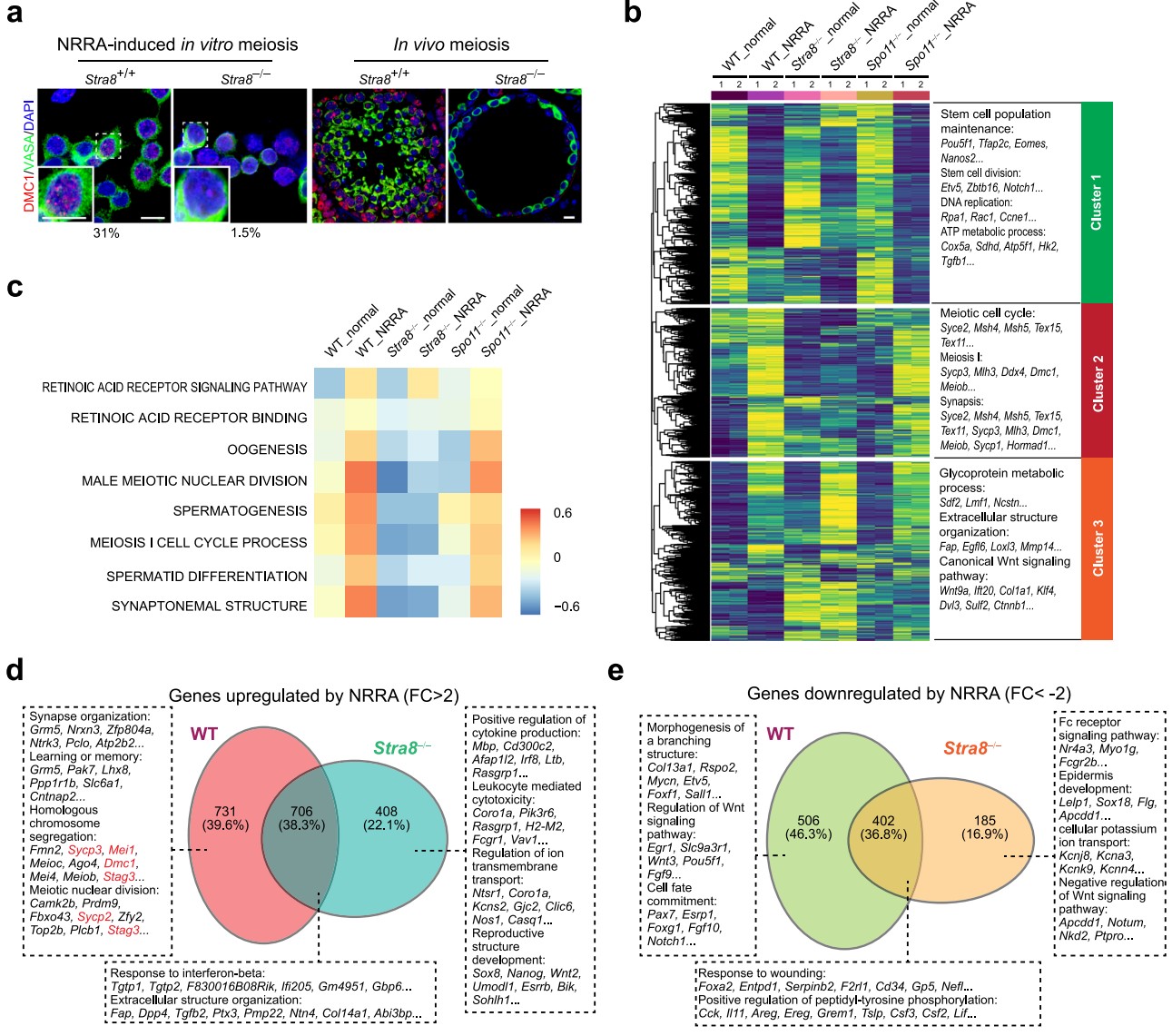

**Fig. 4 *Stra8* is required for NRRA-induced meiotic gene program. a** Immunostaining for DMC1 (red), DDX4 (green), and DAPI (blue) in *Stra8*[+/+] and *Stra8*[−/−] SSC cultures (left panels) and testicular cross sections (right panels). Scale bars, 10 μm. The numbers below the immunofluorescent images indicate the percentage of cells seen with DMC1 staining under different genotypes. **b** (Left) UHC and heatmap of gene expression in SSC cultures with indicated genotypes for 2 days. (Right) Top GO enrichments with representative genes in each cluster. **c** GSVA analysis for indicated genotypes. In the heatmap, rows are defined by the selected gene sets, and columns by consensus scores for each genotype. Group enriched gene sets are highlighted by different color. **d**, **e** Venn plots with top GO enrichments and representative genes for (**d**) NRRA-upregulated and (**e**) NRRA-downregulated genes in *Stra8*[+/+] and *Stra8*[−/−] cultures. Data are representative of (**a**) three independent experiments (SSC cultures under NRRA treatment) or mice from each genotype.

in vitro. These phenotypes are consistent with the phenotype of meiotic initiation arrest of *Stra8*-dificient germ cells in vivo[34,36]. Some germ cells from *Spo11*-dificient culture displayed small areas of SYCP1/SYCP3 co-localization (Fig. 5c, d). However, the typical thread-like co-localization of SYCP1/SYCP3 observed in WT culture was not detected in *Spo11*-dificient culture (Fig. 5c, d), which is consistent with the essential role of Spo11 in synaptonemal complex formation in vivo[4].

**NRRA-induced meiotic DSBs**. Our results show that *Stra8*-dependent and *Spo11*-dependent meiotic DSBs are induced by NRRA in vitro (Figs. 1d, 4a). To further examine the processing of these meiotic DSBs, our spread analysis shows that, similar to in vivo meiosis, single-strand DNA (ssDNA)-binding proteins, SPATA22 and MEIOB[37,38], were recruited onto the meiotic

chromosomes, suggesting that meiotic DSBs formed in vitro were resected into ssDNA before recruiting meiotic recombinases RAD51 and DMC1 for repair (Fig. 6a). Quantification show that similar numbers of SPATA22, MEIOB, RAD51, and DMC1 foci were detected on meiotic chromosomes per germ cell between in vivo meiosis and in vitro meiosis (Fig. 6b). In addition, histone methyltransferase PRDM9 directs meiotic DSBs to be distributed to recombination hotspots[39,40]. PRDM9 expression is induced by NRRA in cultures at both mRNA (Figs. 1h, 2c) and protein levels (Fig. 6c). Consistently, chromatin immunoprecipitation (ChIP) assay for DMC1-associated ssDNA fragments, a direct method to detect recombination hotspots[41], revealed that the DSBs formed in vitro upon NRRA treatment were mapped to the strong hotspots of meiotic recombination in vivo (Fig. 6d, e, Supplementary Fig. 22a) (ref. [42]). Since these recombination hotspots are adjacent

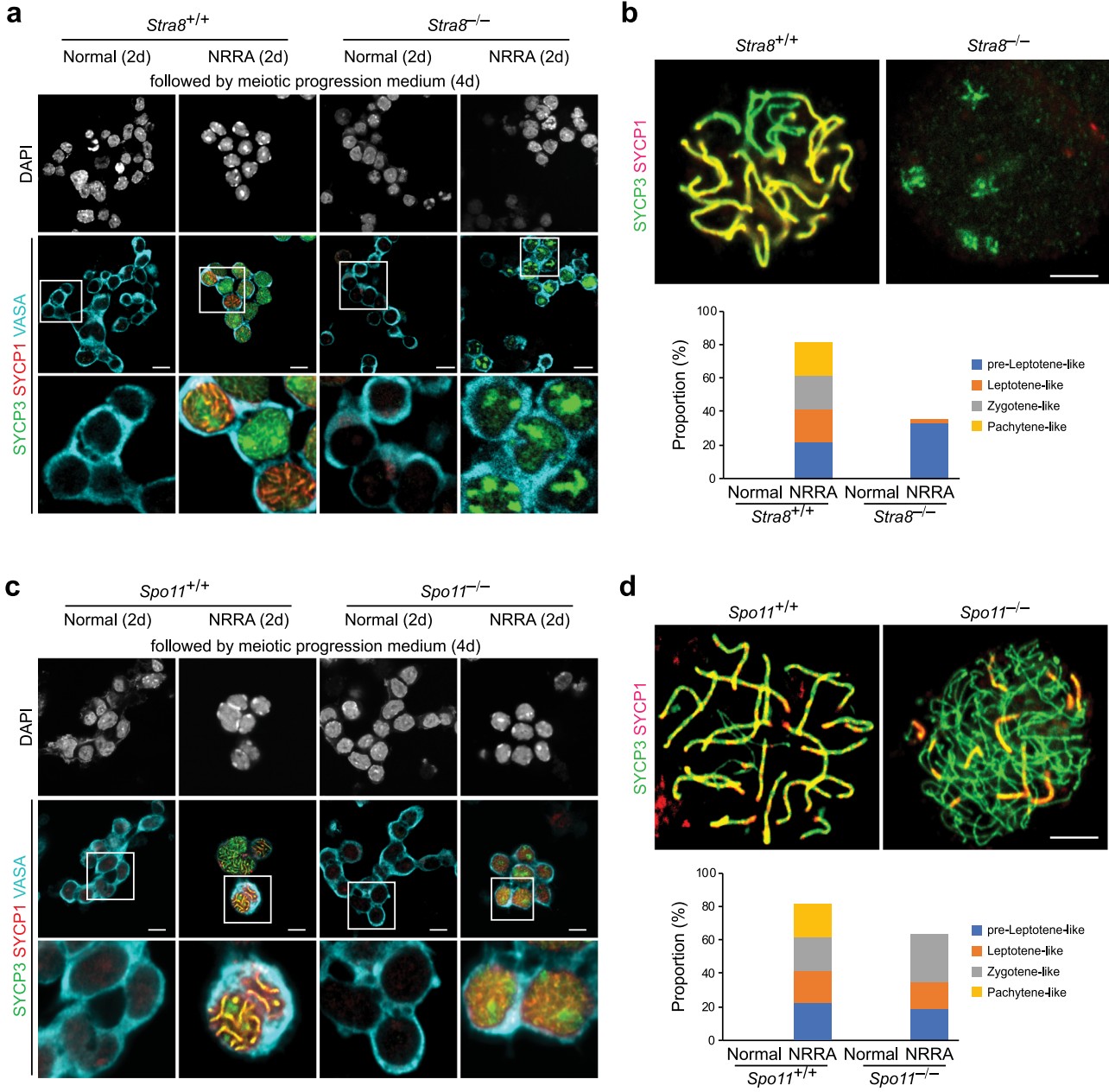

**Fig. 5 Stra8 and Spo11 are required for NRRA-induced meiotic chromosome synapsis. a** DAPI staining and immunostaining for SYCP1 (red), SYCP3 (green), and VASA (cyan) in Stra8$^{+/+}$ and Stra8$^{-/-}$ SSC cultures with indicated treatment. Scale bars, 10 μm. **b** Representative chromosome spreads stained by SYCP1 (red) and SYCP3 (green) from Stra8$^{+/+}$ and Stra8$^{-/-}$ SSC cultures. Percentages of germ cells at the indicated meiotic stages are shown below (n = 50 cells). **c** DAPI staining and immunostaining for SYCP1 (red), SYCP3 (green), and VASA (cyan) in Spo11$^{+/+}$ and Spo11$^{-/-}$ SSC cultures with indicated treatment. Scale bars, 10 μm. **d** Representative chromosome spreads stained by SYCP1 (red) and SYCP3 (green) from Spo11$^{+/+}$ and Spo11$^{-/-}$ SSC cultures. Percentages of germ cells at the indicated meiotic stages are shown below (n = 50 cells).

to the SPO11 cleavage sites (Supplementary Fig. 22b), this data lends further support that NRRA-induced meiotic DSBs in vitro are produced by SPO11 (Fig. 1d).

**Nutrient restriction upregulates a set of TF genes not regulated by RA**. To examine the mechanism of nutrient restriction-induced meiotic initiation, we identified that nutrient restriction-upregulated 120 TF genes, whose expression does not require RA (Fig. 7a, b). To identify those potentially involved in regulating the meiotic gene programs, we examined the expression of these genes in the transcriptomic database for mouse spermatogenesis and found that 30 of them are expressed before meiotic prophase (Fig. 7b, c, Supplementary Fig. 23)[31]. Then, we further

investigated their correlation with meiotic gene expression in the Genotype-Tissue Expression database (GTEx)[43]. Using Dmc1, Sycp3, Hormad1 as three representative meiotic gene, we found that 11 nutrient restriction-upregulated TF genes show strong (Pearson correlation > 0.6) or moderate correlation (Pearson correlation > 0.4) with meiotic gene expression (Fig. 7b, d, Supplementary Fig. 23). We further confirmed that the expression of these 11 TFs in vivo does not require RA (Supplementary Fig. 24)[30]. Notably, these 11 TFs include Sohlh1 and Sox3, two characterized TFs implicated in early meiosis and spermatogenesis. Sohlh1 is basic helix-loop-helix (bHLH) TF, whose deletion results in many tubules lacking meiotic spermatocytes[44]. Recently, Sohlh1 is shown to regulate meiotic gene expression

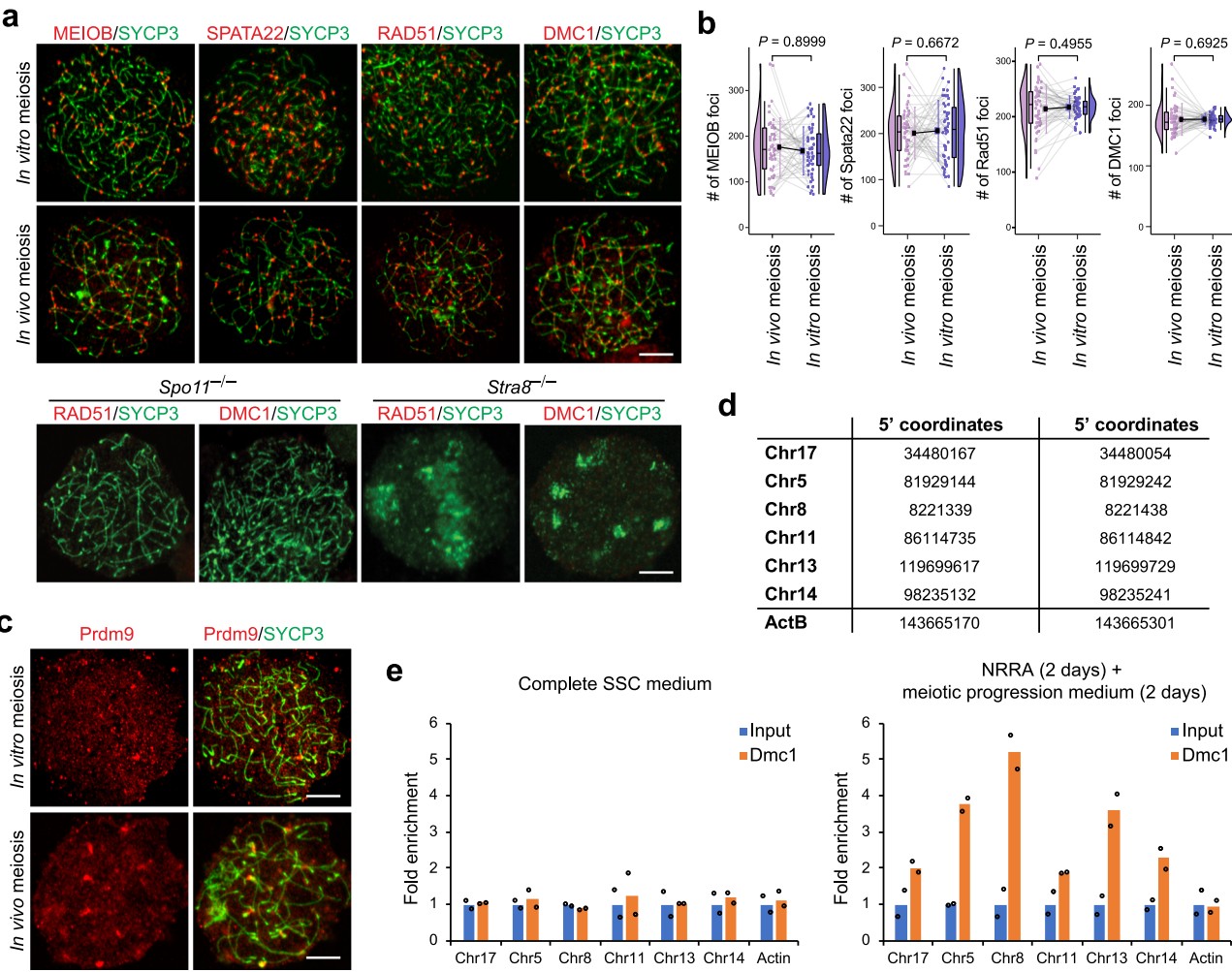

**Fig. 6 Number and distribution of meiotic DSB formation in vitro. a** Representative chromosome spreads stained by RAD51, DMC1, MEIOB, SPATA22, and SYCP3 from juvenile mice (day 14 of age) or SSC culture (wild-type: upper panels; *Stra8*- and *Spo11*-deficient: lower panels) following 2 days of NRRA treatment plus 2 days of treatment in meiotic progression medium. Scale bars, 5 μm. **b** Quantification of foci for zygonema stage. Error bars, mean ± SD. Each solid circle indicates the total number of foci from a single nucleus. Total number of cells quantified from three independent cultures are shown on each graph. n.s. not significant. *n* = 50 cells. Data are provided in the Source Data. **c** Immunofluorescence staining for SYCP3 and PRDM9 on chromosome spreads in testicular germ cells from juvenile mice (day 14 of age) or in cells from SSC culture following 2 days of NRRA treatment plus 2 days of treatment in meiotic progression medium. Scale bars, 5 μm. **d** Chromosome coordinates for previously characterized recombination hotspots (Chr17, Chr5, Chr8, Chr11, Chr13, and Chr14) and *β-actin* gene promoter. **e** Enrichment of the DNA corresponding to recombination hotspots by anti-DMC1 ChIP estimated by qPCR. ChIP was performed from SSC culture following NRRA-induced meiotic initiation and progression. Enrichment of the DNA corresponding to genomic regions on Chr17, Chr5, Chr8, Chr11, Chr13, Chr14, and the *β-actin* gene was estimated by qPCR. *n* = 2 (duplicate PCR reaction). Data are representative of (**c**) three and (**e**) two independent experiments. *P* value by (**b**) student *t* test.

(*Sycp1*, *Sycp3*) by directly binding to their proximal promoters[45]. Sox3 is expressed exclusively in spermatogonia committed to differentiation[46], and loss of Sox3 impairs early meiosis, including a downregulation of *Sycp3* expression in young adult testes[47]. Thus, these data suggest that nutrient restriction induces a distinct network of TFs to complement RA signaling in the meiotic gene activation (Fig. 8).

## Discussion
The transition from mitotic to meiotic cell cycle is a defining feature of germ cells. Although the molecular machinery and chromosomal dynamics underlying this process is often conserved from yeasts to mammals, the signal to induce meiosis appears different: nutrient restriction induces meiotic initiation in the yeast system, whereas RA, a chordate morphogen, and its signaling have been the primary focus in mammals. However, a gap of knowledge exists in mammals in that RA signaling is

necessary but not sufficient to induce meiotic initiation, suggesting that other signal(s) is required. Here, by using primary spermatogonial culture as a platform, our study establishes a role for nutrient restriction in inducing meiotic initiation in mammalian germ cells, in that nutrient restriction in combination with RA is sufficient to robustly induce meiotic initiation that faithfully recapitulates the transcriptomic, chromosomal, and genetic features of in vivo meiosis. Neither nutrient restriction nor RA alone exhibits this effect. Moreover, we have identified 11 TF genes upregulated by nutrient restriction, but not RA, that are associated with early meiosis. Together, our study suggests a conserved role for nutrient-sensing pathway in inducing meiotic initiation in mammals.

Homologous recombination is the crux of the meiosis. To initiate recombination, hundreds of DNA DSBs (200–300 in mice; ~150 in human) must be deliberately formed[48]. Directed by PRDM9, these DSBs are distributed in particular regions along

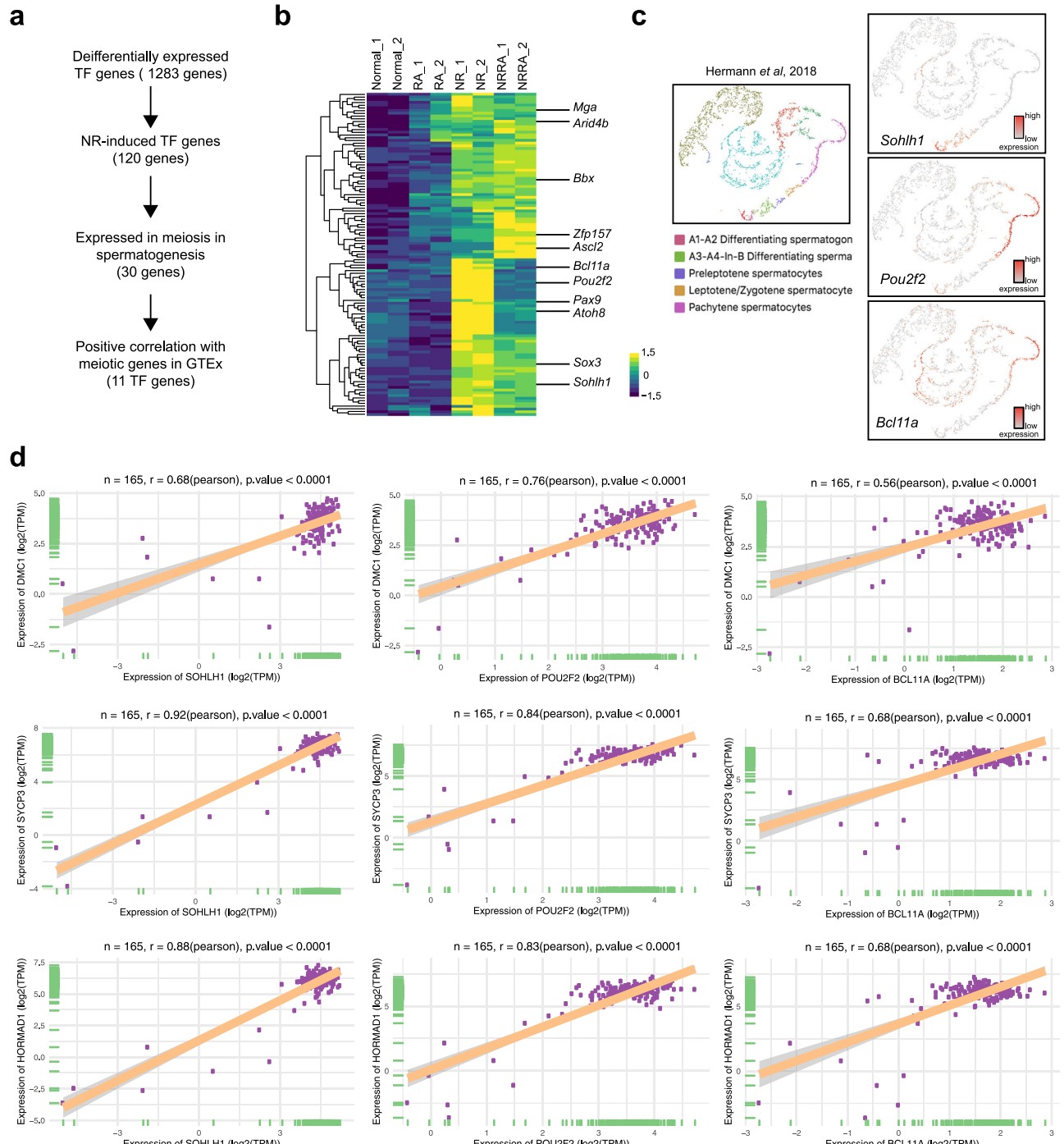

**Fig. 7 Nutrient restriction induces a network of TF genes involved in early meiosis. a** A workflow showing filters to identify genes of interest. **b** Heatmap of 120 TF genes upregulated by nutrient restriction. 11 identified TF genes involved in early meiosis are shown on the right. **c** Gene expression patterns of 3 representative TF genes during early spermatogenesis from a published scRNA-seq database[31]. **d** GTEx database showing correlation of representative TF genes (*Sohlh1, Pou2f2,* and *Bcl11a*) with selected meiotic genes in 165 testis samples. The linear regression curve is demonstrated. 95% confidence interval (shaded area) is shown in each panel. The range of confidence interval is provided in the Source Data. Two tailed *t*-statistic *P* value and coefficient (*R*) of Pearson's correlation is shown on the top.

the chromosomes in mammals[39,40]. The formation of DSBs, a most dangerous form of DNA damage to somatic cells, at this magnitude is not easy: as an example, while 1 Gy of ionizing irradiation produces merely ~20 DSBs, 2 Gy often causes cell death. In germ cells, the production of DSBs is catalyzed by a meiosis-specific DNA topoisomerase-like enzyme, Spo11, which is conserved from yeasts to mammals. However, the transcriptional regulation of *Spo11* gene is not well understood. Although

STRA8 and MEIOSIN have recently been shown to directly bind to the regulatory sequence of *Spo11*, they are not sufficient to activate *Spo11* transcription[17]. Our study demonstrates that NRRA is sufficient to induce Spo11-catalyzed meiotic DSBs in vitro. We show that their numbers and distribution are comparable to those formed during in vivo meiosis. Thus, our study suggests that, like in yeasts, nutrient-sensing pathway is involved in the transcriptional regulation of *Spo11*.

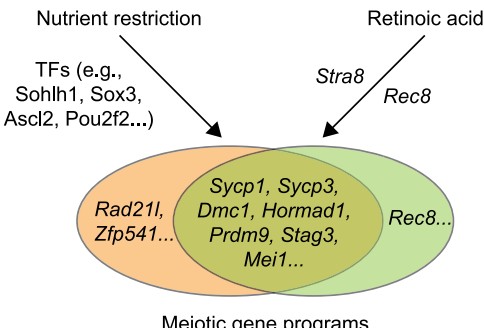

**Fig. 8 Schematics of nutrient restriction in combination with retinoic acid in inducing meiotic gene programs.** Nutrient restriction alone is sufficient to activate a class of meiotic genes that do not require retinoic acid (e.g., *Rad21l*). Retinoic acid alone is sufficient to activate *Rec8*, an essential component of cohesion, and *Stra8*. Nutrient restriction and retinoic acid synergistically induce key meiotic genes for chromosome synapsis and recombination.

The NRRA condition to induce meiotic DSBs and initiation requires *Stra8* (Fig. 4). Recently, we show that STRA8 acts as a suppressor of autophagy[24]. Thus, the NRRA condition can be viewed as applying metabolic stress to autophagy-deficient cells. Interestingly, a past study in somatic cells have shown that, in response to metabolic stress, autophagy-deficient cells exhibit increased cell sizes and chromosome instability, including profound DNA DSB formation[49]. Thus, this information together suggests a crucial role for the interplay between autophagy and nutrient-sensing pathway in regulating chromosome remodeling and cell fate decision.

The nature of the nutrient restriction condition—that is, restriction of which nutrient type(s) is necessary for meiotic induction—remains to be further defined. In yeasts, nitrogen starvation triggers meiosis. It appears that some yeast factors that respond to nitrogen starvation have become sensitive to amino acid starvation in mammalian cells. For instance, Dhh1 responds to nitrogen starvation in yeasts, while its mammalian homolog, Ddx6, responds to amino acid starvation[50]. Interestingly, past studies that characterized the fluid composition inside seminiferous tubules and showed that it is low in most amino acids compared to blood plasma[51]. Consistently, EBSS used in our nutrient restriction medium applies amino acid starvation to cells. Thus, we speculate that amino acid restriction is a critical nutrient signal for mammalian meiotic initiation.

Nutrient restriction is clearly a metabolic cue. Our scRNA-seq revealed that, while genes involved in the glycolytic pathways were downregulated, genes involved in mitochondrial function pathways were upregulated by NRRA (Supplementary Fig. 25), which mirrors the transition from glycolysis to mitochondrial biogenesis and oxidative phosphorylation during in vivo spermatogonial differentiation[52]. Then, how do germ cells achieve this metabolic switch in vivo? A model has been proposed for the differential microenvironments resided by undifferentiated and differentiating spermatogonia in the testis, which are influenced by the distances to blood supply and oxygen[52]. Moreover, inside seminiferous tubules, the blood–testis barrier (BTB) is a dynamic ultrastructure that restricts paracellular flow of biomolecules or substances to the adluminal compartment of the tubule, thus creating a nutrient-restricted microenvironment in which meiosis takes place[53–55]. BTB establishment by expression of its major components (e.g., Cx43, Claudin-11, and Occludin) commences in mouse testes from ~6 days of age, concomitant with the first wave of meiotic initiation[56–59]. During spermatogenesis, the dynamic assembly and disassembly of the BTB required to

transport germ cells accompanies their meiotic initiation[60]. Functionally, BTB component is required for meiotic initiation, as genetic deletion of Cx43, a gap junction protein, results in meiotic initiation arrest, including that in the first wave of spermatogenesis, and, intriguingly, a downregulation of the early meiosis-associated TF genes that we show whose expression requires nutrient restriction but not RA (*Ascl2*, *Pou2f2*, *Sohlh1*, and *Sox3*) (ref. [56,61–63]). Although there are genetic mouse models in which impairment of BTB integrity appears not to affect meiotic initiation (reviewed in ref. [64]), it is possible that individual BTB protein components play distinct roles in controlling substance flow across the BTB and that some component(s), e.g., Cx43, may have a specific role in generating the nutrient signal for meiotic initiation. For female meiosis, we speculate that the germ cell nest formed in fetal ovaries (ref. [65]) may provide a similar nutrient restriction function to induce meiotic initiation in primordial germ cells.

In addition to the development of SSC culture from multiple mammalian species, mounting efforts have been directed to derive stem cells with germline potential from pluripotent stem cell sources, such as embryonic stem cells and induced pluripotent stem cells (reviewed in ref. [21]). By our study provides a relatively simple and easily reproducible approach, which not only provides a method to induce meiosis but also understand the molecular mechanism underlying meiotic initiation from the perspective of nutrient-sensing pathway.

## Methods

**Animals.** All animal experiments were conducted according to the approved protocol by the Institutional Animal Care and Use Committee at the University of Kansas Medical Center in strict accordance with its regulatory and ethical guidelines. All animals were housed in a specified pathogen-free facility with a 12 h light/dark cycle. All animals had access to food and water ad libitum. *Stra8*-deficient mice were obtained from Jackson Laboratory (Jax stock number: 023805). *Spo11*-deficient mice were obtained from Keeney/Jasin lab and are available at Jax (Jax stock number: 019117). CD1 and DBA/2 mice were raised under specific-pathogen-free (SPF) condition.

**Primary culture of mouse spermatogonia (SSC culture).** We generated primary culture of mouse spermatogonia (SSC culture) in both CD1 inbred and C57BL/6XDBA/2 F1 hybrid backgrounds[18]. To establish SSC culture, testes from 6-day-old mouse pups were dissociated by a two-step enzymatic digestion procedure. Briefly, after removal of the tunica albuginea, testes were cut into small pieces and transferred to a 15 ml tube containing 4 ml of collagenase IV solution (1 mg/ml in HBSS). Tissues were incubated in 37 °C water bath for 20 min when tissues are separated. After washing the tubule fragments with 4 ml of HBSS and centrifuging at 300 × *g* for 5 min, the tubule fragments were incubated in 4 ml of 0.05% trypsin-EDTA solution in 37 °C water bath for 5 min. Testicular cells were cultured on 0.1% gelatin-coated plates with complete SSC medium overnight. Then, non-adherent cells (germ cells) were collected, while adherent cells (somatic cells, e.g., Sertoli cells) were discarded (differential plating to enrich stem cells). Collected non-adherent cells (germ cells) were transfer on to MEF feeder cells for expansion and long-term culture. MEFs were purchased from Thermo Fisher (cat# A34180). According to manufacture website, MEF cells are collected from outbred CF1 mice, mitotically arrested by irradiation. For indicated treatments to induce meiotic initiation, $1.0 \times 10^6$ cells from SSC culture were plated on 60-mm dishes coated with MEF feeder cells and treated for 48 h. At meiotic initiation, meiotic progression was induced by the "meiotic progression" medium supplemented with melatonin, FSH, TGFβ, BPE, and DHT. Compositions for all media used in this study were summarized in Supplementary Table 1.

**RNA isolation and qPCR.** Total RNA was isolated from collected cells using TRIzol™ (Invitrogen,15596018). The RT reaction was carried out with SuperScript II First-Strand Synthesis Kit (Invitrogen,18080-051). qPCR was performed with gene specific primers that were listed in Supplementary Table 2. qPCR with amplified cDNAs was performed using the Power SYBR Green Master Mix (Applied Biosystems) on Applied Biosystems Quant Studio 5.

**Immunofluorescence (IF) analysis.** For the IF staining, cultured cells were fixed with 4% paraformaldehyde contained 0.1% Triton X-100 for 10 min at room temperature. Then cells were washed with PBS. Blocking was performed using 5% BSA for 1 h at room temperature. The primary antibodies were added and incubated for overnight at 4 °C. After washed in PBS, the secondary antibodies were

added and incubated for 1 h at room temperature. Images were captured using Nikon A1R confocal microscope and were processed using Nikon NIS Elements and Adobe Photoshop.

**Antibodies for immunofluorescence**. Primary antibodies used were: DMC1 (Sigma, HPA001232) 1:200 dilution; SYCP3 (Santa Cruz Biotechnology, SC-74569) 1:200 dilution; SPATA22 (Proteintech, 16989-1-AP) 1:100 dilution; MEIOB (Gift from Jeremy P. Wang's lab) 1:100 dilution; RAD51 (Millipore, PC-130) 1:200 dilution; γH2AX (Millipore, 05-636) 1:800 dilution; DDX4 (Abcam, Ab13840) 1:200 dilution; GFRA1 (R&D, AF560) 1:200 dilution; CDH1 (BD, 610181) 1:200 dilution. Secondary antibodies, donkey anti-mouse, donkey anti-rabbit, or donkey anti-goat antibodies conjugated to AlexaFluor-488, AlexaFluor-546 or AlexaFluor-647, were purchased from Thermo Fisher and used at 1:500 dilution.

**Chromosome spreads**. Testis were dissociated after removing the tunica albuginea in 15 mL tubes with 5 mL TIM buffer (104 mM NaCl, 45 mM KCl, 1.2 mM MgSO₄, 0.6 mM KH₂PO₄, 6.0 mM sodium lactate, 1.0 mM sodium pyruvate, and 0.1% glucose) containing 1 mg/ml collagenase IV. Tissue was left shaking for 1 h at room temperature. After incubation, TIM buffer was removed after centrifugation for 5 min at $300 \times g$ at room temperature. A 1 ml tip was used to disperse the tissue further by pipetting up and down for 5 min. Single-cell suspension was resuspended in 500 μl TIM. 75 mM sucrose solution was added to tube. 200 μl cell mix was transfer to Superfrost plus glass slides (Fisher Scientific) coated with a thin layer of 1% paraformaldehyde (PFA), and half-dried in a closed slide box for 1 h, then dried with half-open lid for 2 h at room temperature. Slides were washed with 0.4% PhotoFlow (Nikon), air-dried and stored in −80 °C.

**Western blot**. Total protein was isolated in RIPA buffer contained with 1 mM PMSF (Sigma) and protease inhibitor cocktail (Sigma P8340). Lysates were removed by centrifugation at $12,000 \times g$ for 15 min at 4 °C, and protein concentrations in supernatants were determined (DC protein assay; BioRad). Equal amount of protein from each sample was mixed with LDS sample buffer (Invitrogen) plus sample reducing agent (Invitrogen), and denatured for 10 min at 70 °C. Proteins were resolved in 4–12% Bis-Tris gels (Thermo Fisher) and transferred to PVDF membranes. Blots were probed with antibodies overnight at 4 °C, washed and reacted with secondary antibody. Detection was performed with the Clarity ECL Western Blotting Substrate (BioRad).

**Chromatin immunoprecipitation (ChIP)**. Testicular cells or collected cells were dissociated after remove the tunica albuginea and fixed for 10 min in 1% formaldehyde. Followed by quench the tissue with homogenized and washing in PBS twice. Cells were lysed in 1 ml of the lysis buffer (from EZ-ChIP kit, Millipore,17-371RF) and the chromatin was sheared to ~1000 bp by sonication using a Diagenode pico bioruptor. The sample was pre-cleared with Protein G beads and 10 μl of pre-cleared chromatin was saved and referred to as input. The rest of the chromatin was incubated with DMC1 antibody or preimmune IgG overnight at 4 °C, followed by a 2 h incubation with Protein Agarose G beads. Following washing, the chromatin was eluted from the agarose beads by using 1% SDS, 0.1M NaHCO₃ (pH 9.0) at 65 °C and crosslinking was reversed at 65 °C overnight. DNA was deproteinized for 2 h at 45 °C and DNA was purified. Enrichment of the characterized hotspots was evaluated by qPCR analysis using previously reported primers (Supplementary Table 2) (ref. [42]).

**RNA-sequencing (RNA-seq)**. The stranded mRNA-Seq was performed using the Illumina NovaSeq 6000 Sequencing System at the University of Kansas Medical Center Genomics Core (Kansas City, KS). Quality control on RNA submissions was completed using the Agilent TapeStation 4200 using the RNA ScreenTape Assay kit (Agilent Technologies 5067–5576). Total RNA (1 μg) was used to initiate the library preparation protocol. The total RNA fraction was processed by oligo dT bead capture of mRNA, fragmentation, reverse transcription (RT) into cDNA, end repair of cDNA, ligation with the appropriate Unique Dual Index (UDI) adaptors, strand selection and library amplification by PCR using the Universal Plus mRNA-seq with NuQuant library preparation kit (Tecan Genomics 0520-A01). Library validation was performed using the D1000 ScreenTape Assay kit (Agilent Technologies 5067–5582) on the Agilent TapeStation 4200. Concentration of each library was determined with the NuQuant module of the library prep kit using a Qubit 4 Fluorometer (Thermo Fisher/Invitrogen), libraries were pooled based on equal molar amounts and the multiplexed pool was quantitated, in triplicate, using the Roche Lightcycler96 with FastStart Essential DNA Green Master (Roche 06402712001) and KAPA Library Quant (Illumina) DNA Standards 1–6 (KAPA Biosystems KK4903). Using the qPCR results, the RNA-Seq library pool was adjusted to 2.125 nM for multiplexed sequencing. Pooled libraries were denatured with 0.2N NaOH (0.04 N final concentration) and neutralized with 400 mM Tris-HCl pH 8.0. A dilution of the pooled libraries to 425 pM is performed in the sample tube, on instrument, followed by onboard clonal clustering of the patterned flow cell using the NovaSeq 6000 S1 Reagent Kit (200 cycle) (Illumina 20012864). A 2 × 101 cycle sequencing profile with dual index reads is completed using the following sequence profile: Read 1—101 cycles × Index Read 1—8 cycles × Index Read 2—8 cycles × Read 2—101 cycles. Following collection, sequence data is converted from.bcl file format to fastq file format using bcl2fastq software and de-multiplexed into individual sequences for data distribution using a secure FTP site or Illumina BaseSpace for further downstream analysis.

**Data analysis of RNA-seq**. Bulk RNA-seq data analysis was performed using the R software (version 3.6). DEGs were calculated by DEseq2 (ref. [66]). The heatmap was generated using the pheatmap. GO analysis of genes were analyzed by clusterProfiler package[67]. GSVA heatmap was generated by R package, GSVA (ref. [68]).

**Single-cell RNA-seq (scRNA-seq)**. Treated cells were digested by collagenase IV for 10 min at room temperature. The digestion was then stopped by media. The cells were pelleted by centrifugation at $300 \times g$ for 5 min. And then, supernatant was removed, and the cell pellet was washed with PBS twice. Single cells were obtained by filtering through 40 μm strainers. Cell number was counted using Countess II FL automated cell counter (Invitrogen). The 10× Genomics Single-Cell 3′ Expression library preparation is performed using the 10× Genomics Chromium Controller. The cells prepared from disassociated tissue or tissue culture are validated for viability and cell concentration using the Countess II FL Automated Cell Counter (Life Technologies) targeting ≥75% cell viability. If debris or cell clumping is present in the cell suspension, the preparation is filtered through a FLOWMI Cell Strainer, 40 μm (Thermo Fisher 50-136-7555) to yield a homogenous single-cell suspension. Cell counts are redetermined by using the Countess ll FL and adjusted to ~1000 cells/μl by low speed centrifugation at 4 °C and resuspended in 1× PBS without calcium or magnesium (Thermo Fisher MT21040CV) supplemented with 0.04% BSA to prepare cells for emulsification. The cell emulsification is performed with the 10× Chromium Controller using the Chromium Next GEM Single-Cell 3′ GEM Library & Gel Bead Kit v3.1 (10× Genomics 1000120) and Chromium Next GEM Chip G Single-Cell Kit (10× Genomics 1000127). Well 1 of Chip G is loaded with the RT Master Mix + cell suspension containing ~16,000 cells to target 10,000 emulsified cells at ~65% efficiency of emulsification. Well 2 of Chip G is filled with 50 μl of the Next GEM GEL Beads. Well 3 of Chip G is filled with 45 μl Partitioning Oil. Any unused wells are filled with 50% glycerol at a volume designated for the well number. A gasket is applied to the Chip G and the loading cassette and inserted into the Chromium Controller for GEM creation Using the Chromium Single-Cell G run program. Emulsified GEMs are recovered from each well and transferred to 200 μl strip tubes. The RT reaction to generate 10× barcoded single stranded cDNA, in the single-cell containing GEMs, is performed using an Eppendorf MasterCycler Pro thermal cycler. Post GEM-RT Cleanup is conducted by breaking the GEMS with 10X Recovery Agent to separate the aqueous phase from the Recovery Agent and Partitioning oil. A cleanup of the sscDNA containing aqueous phase is completed using Dynabeads MyOne Silane beads (Life Technologies 37002D). The second strand cDNA amplification is performed using the cDNA Amplification Mix on the Eppendorf MasterCycler Pro. The 3′ gene expression library construction is initiated with fragmentation, end repair and A-tailing of the dscDNA followed by adapter ligation and a sample index PCR using the Illumina compatible indexed adapters in the Chromium i7 Multiplex kit (10× Genomics 120262). Validation of the single-cell library is conducted using the Agilent Tapestation 4200 ScreenTape assay (Agilent 5067-5576). Single-cell library quantification is completed using a Roche LightCycle96 using FastStart Essential Green Master (Roche 06402712001) and KAPA Library Quant (Illumina) DNA Standards (KAPA KK4903). Library concentrations are adjusted to 3 nm and pooled. The library pool is diluted to 1.00 nm for a final clustering concentration of 200pM on a NovaSeq6000. The sequencing was performed using a NovaSeq6000 100 cycle Reagent Kit (Illumina 20012865) with an asymmetrical sequencing profile (read 1—28 cycle: i7 index read—8 cycle: i5 index read—0 cycle: read 2—94 cycles). Bcl2fastq conversion and demultiplexing is performed using the 10X Genomics Cell Ranger and Loupe Browser software suite.

**Analysis of single-cell RNA-seq data**. Cell clustering was performed by Seurat (https://satijalab.org/seurat/, R package, v3.1) (ref. [69]). Seurat object was created first. Then, we discarded low-quality cells, in which less than 200 genes were detected. Genes expressed in less than three cells were filtered out. We also filtered cells that have lower than 2000 genes and that contain mitochondrial genome higher than 10% of mapped reads. We then used LogNormalize, a global-scaling normalization method, which normalized gene expression measurements by the total expression per cell, followed by multiplication of the result by a default scale factor (10,000) and subsequent log-transformation. After filtering and normalization, 18,088 cells were selected for further analysis. Gene expression was calculated by centering expression across all cells in the cohort using the "ScaleData()" function in Seurat. Identification of top 6000 variable genes, PCA (ref. [70]) (30 significant PCs determined by a scree plot), JackStraw (https://CRAN.R-project.org/package=jackstraw) (ref. [71]), and SNN-Cliqinspired clustering were performed in Seurat to generate t-distributed stochastic neighbor embedding (t-SNE) visualizations[72]. Cluster definition was performed by specific marker genes and the "FindClusters()" function in Seurat. Clusters 0–3 as germ cells based on germ cell marker gene, *Ddx4*, expression and cells in Cluster 4 as mouse embryonic fibroblast (MEF) based on fibroblast marker gene, *S100a4*, expression. Gene expression analysis for different clusters were performed by using the Monocle (http://cole-trapnell-lab.github.io/monocle-release/, R package, v2.12) under the

default settings[73]. For the gene expression analysis, we used a published code from a published study with modifications[74]. Pseudo-time lineage reconstruction was performed by using diffusion map in the Destiny R package on the set of top 6000 variable genes from Seurat analysis[75]. Trajectories were calculated by using the diffusion map results as an input data and using the Slingshot R package[76]. Pheatmap package was used for heatmap plotting and clustering[77]. GO analysis of genes were analyzed by using the ClusterProfiler package.

**Relationship analysis of cell clusters in in vitro meiosis and in vivo meiosis**. For in vivo meiosis, we used a published scRNA-seq dataset on mouse spermatogenesis[31]. To analyze the relationship between the cell clusters identified by scRNA-seq during in vitro meiosis with the cell clusters during in vivo meiosis, we used a method from a published study with modifications[78]. Briefly, we calculated the mean expressions of genes in each cell cluster as "pseudo-bulks" by using Monocle2. Next, "ComBat" function from the R/Bioconductor sva package was applied to regress-out the technical effect[79]. Then, the first two PCs of the corrected expression were calculated by using the "prcomp" R function. The plot for PCA analysis was generated by ggplot2[80].

**Statistical analysis**. All experiments were replicated at least three times independently. Different mice, tissues or cells were used during each experimental replicate. Quantitative data from the experimental replicates were pooled and are presented as the mean ± SEM or mean ± SD as indicated in the figure legend. Compiled data were analyzed by Student's *t* test and two-way ANOVA test.

**Reporting summary**. Further information on research design is available in the Nature Research Reporting Summary linked to this article.

## Data availability

The authors declare that all data supporting the findings of this study are available within the article and its Supplementary Information files or from the corresponding author upon reasonable request. FastQ files of RNA-seq and single-cell RNA-seq are available on Gene Expression Omnibus (GEO) database under accession code GSE153300. Source data are provided with this paper.

## Code availability

Source code of the analysis is publicly available on GitHub at this address: https://github.com/iamzhangxiaoyu/scRNA-seq and are available on Zenodo: https://doi.org/10.5281/zenodo.4535405.

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

## Acknowledgements

We thank Leslie Heckert (KUMC) for helpful input on this study, Kyle Orwig (Magee-Women Research Institute & Foundation/University of Pittsburgh) for suggestion on primary mouse spermatogonial stem cell culture, Petko Petkov (The Jackson Laboratory) for Prdm9 antibody, Scott Keeney, and Maria Jasin (Memorial Sloan Kettering Cancer Center) for Spo11-deficient mice, and Jeremy Wang (University of Pennsylvania) for MEIOB antibody and insightful discussions. We thank Jing Huang at Smith IDDRC Histology Services for histological sections. We thank Clark Bloomer and Rosanne Skinner at the University of Kansas Medical Center Genomics Core. We also thank Dr. Jianming Zeng (University of Macau) and all the members of his bioinformatics team, biotrainee, for generously sharing their experience and codes. This work was supported by the National Institutes of Health (NIH) grants (R21HD-087741; R01HD-103888) to N.W. X.Z. is a recipient of KUMC Biomedical Research Training Program (BRTP) postdoctoral fellowship. N.W. has been supported by a startup fund from the Department of Molecular and Integrative Physiology at KUMC. The University of Kansas Medical Center Genomics Core is supported by Kansas Intellectual and Developmental Disabilities Research Center (NIH U54 HD 090216), the Molecular Regulation of Cell Development and Differentiation—COBRE (P30 GM122731-03)—the NIH S10 High-End Instrumentation Grant (NIH S10OD021743) and the Frontiers CTSA grant (UL1TR002366).

## Author contributions

X.Z. performed all the experiments. S.G. assisted the analyses of RNA-seq and scRNA-seq data. X.Z. and N.W. designed the experiments and analyzed data. N.W. and X.Z. wrote the paper.

## Competing interests

The authors declare no competing interests.
