## [Peer Review File · Nature Communications]

Reviewers' comments:

Reviewer #1 (Remarks to the Author):

Summary

In this manuscript, Zhang and colleagues use an ex vivo mouse primary spermatogonia culture system to examine factors associated with meiotic initiation in the male germline. The authors report that with dual treatment of Earle's Balanced Salt Solution ("starvation" media) and Retinoic Acid they observe hallmarks of meiotic entry, including formation of SPO11-induced DSBs, expression of meiosis-related genes, and cytological evidence of meiotic chromosomes. Importantly, the authors show genetically that formation of the DSBs are dependent on the presence of Stra8 as well. Using single-cell RNA-seq, Zhang et al. also compare their culture system to published single-cell RNA-seq datasets of in vivo male germ cells and show that their cultures are comprised of cells with gene expression profiles similar to undifferentiated spermatogonia all the way through to pachytene-stage spermatocytes. Finally, the authors identify 11 candidate transcription factors which are upregulated in the nutrient-restricted condition (independent of RA) and hypothesize that these are factors required for meiotic initiation.

Overall, this manuscript performs many genomic analyses of cells produced in the culture system, and makes an effort to validate the ex vivo findings using relevant in vivo datasets and genetics. While it is apparent that NRRA is inducing meiotic initiation ex vivo, the evidence for the same thing occurring in vivo is not nearly as convincing. One important issue with the manuscript is that there are a number of places where there is unclear/insufficient rationale for experiments and interpretation of data. This paper was clearly drafted originally for a short report format, but I believe that this should be expanded into a full article format so as to clarify the presentation. In some cases, it is unclear to me whether problematic portions are the result of insufficient/unrigorous experimentation, or just brevity. Below are specific comments.

Major comments:

- The specific nature of the Earle's Balanced Salt Solution-based nutrient restriction is unclear. What nutrients are being restricted when EBSS is used? Why was EBSS chosen as the nutrient restriction agent? Would any agent which caused nutrient restriction be expected to yield similar results?
- The authors do not discuss the extensive literature on in vitro spermatogenesis that has emerged in recent years, including differentiation of pre-spermatogonial cells. It might be useful to compare the efficiencies and the extent to which germ cells develop in these other systems.
- Antibody staining in Fig. 1B-D should be done on meiotic chromosome preparations, not whole cells. If these cells are really undergoing meiosis, then the chromosomes should be assuming a true meiotic appearance. Later figures have chromosome spreads. Nevertheless, there are no examples of pachytene spermatocytes. Is there no progression to this stage with further culture?
- Lines 106-111. Just because CD1 mice had similar results doesn't mean one can confidently say that "the sufficient role of NRRA in inducing meiotic gene program is independent of genetic backgrounds." Only two backgrounds were tested, the first being an F1 hybrid.
- Clarification about the MEF feeder layer is important. I assume the primary MEFs were rendered mitotically inactive? If so, how was this done? Senescent cells are not entirely inert, they secrete factors into media; therefore if the protocol requires this feeder layer, it implies that the MEFs are contributing to the culture conditions. The methods should clarify the mouse strain used to generate the MEFs.

- Lines 132-133. Elaborate on "...a medium that is supposedly to promote meiotic progression for an additional 1 and 2 days."
- Figure S22 purports to show early pachytene-like spermatocytes. That is clearly not the case; they look like late leptoneuma or maybe zygonema at best. It would be a good idea to comment at some point on the timing of progression from pre-pubertal gonial to putative pachytene spermatocytes in this system compared to in vivo. It seems like 2-4 days as reported here is extremely rapid.
- Lines 231-235. No convincing data, or insufficient data, is shown to support the conclusion that the NRRA-induced meiosis has the same recombination hotspots as in vivo. Fig. S22b does not match the legend, which says that two cell types were compared. The exact locus coordinates were not given. This dataset is not acceptable for publication as presented or discussed.
- Along the lines of MEFs, Cluster 4 (comprising of the MEFs) is absent in Figure 2D+E. I think readers would be interested in seeing Cluster 4 represented in those figures.
- The last sentence of the paper mentions that the in vitro platform can facilitate both the study of meiotic initiation and *the production of haploid gametes in culture*. What happens to the cells after the completion the treatments described? Do they continue onwards in the meiotic program?
- Lines 67-68: "We established primary spermatogonial culture by using neonatal mouse testicular cells in C57BL/6 X DBA/2 F1 hybrid background." These were from whole testes, but it is mentioned in the methods that "stem cell enrichment" was performed. What is that? How enriched? Is it possible that there are testis somatic cells in the cultures, in addition to the MEF feeders? This needs to be validated in detail.
- Last paragraph of text. The authors claim that their results are consistent with the in vivo situation, in which the BTB causes a nutrient starved domain that triggers meiosis. However, the first wave of spermatogenesis in mice occurs before the BTB is formed, beginning 15 days postpartum. This begs the question of whether the results here, which involve neonatal germ cells, does not accurately represent the situation of meiotic entry in adults.

Minor comments:

Line 56: The mesonephric ducts are not part of the ovary.

Page 4, line 116: "Unsupervised UHC" is redundant

Page 5, line 132: "supposedly" should be "supposed"

Page 8, line 223: "sing-strand" should be "single-strand"

- Lines 15-17 of the Abstract are confusing: "nutrient restriction is both necessary and sufficient to robustly induce Spo11-dependent meiotic DNA DSBs and Stra8-dependent meiotic gene programs" but then continues to say "with RA..." Is NR necessary and sufficient for meiotic induction or is RA also needed? If RA is also needed, then it doesn't seem like NR is sufficient for meiotic induction...
- Page 8, lines 243-245: This sentence describing a result is also very confusing.; "Using Dmc1, Sycp3, Hormad1 as three representatives, we found that 11 of them..."
- It might be worthwhile to comment on how the timing of your NRRA treatment conditions compares to the timing/number of days it takes for meiotic initiation in vivo. Is the timing comparable? Or is ex vivo initiation faster?

- Page 7, lines 208-212: This sentence related to the interpretation of the data as currently written is very confusing and must be fixed: "...in contrast to WT and Spo11-deficient cultures, despite activation of retinoid acid receptor signaling pathway, gametogenesis- and meiosis-related pathways were not activated in Stra8-deficient culture upon NRRA treatment."
- I do not believe there was any mention of what the percentages below the immunofluorescent images indicate. I am assuming it means the percentage of cells seen with the relevant staining under different conditions (?), but it should be stated in figure legends/where appropriate.
- Like Supplemental Table S9, Table S8 should include the company, catalog and lot numbers associated with the media components. This is important because tissue culture media components can be highly variable between companies and lot numbers.
- In acknowledgements, it says that Spo11 mutant mice were obtained from Jasin/Keeney, but in the Methods it says they were obtained from JAX. This (and several other poor descriptions throughout the paper) concerns me in general about the rigor of the research.

Reviewer #2 (Remarks to the Author):

Comment on Zhang et al.

The authors investigated the signals to initiate meiosis in mouse. They performed scRNA-seq analyses using primary spermatogonial culture to see how different culture conditions affected on transcriptome change for spermatogonial differentiation and activation of meiotic genes. They found that transcriptome in primary spermatogonial cells was different under nutrient restriction (NR), retinoic acid (RA) exposure, and combination of both NR and RA compared to untreated control. The authors claimed that nutrient restriction was necessary and sufficient for induction of Spo11-dependent meiotic DSB program and Stra8-dependent meiotic gene program, which led them to the conclusion that nutrient restriction is the conserved mechanism for meiotic initiation as observed in yeast. Their scRNA-seq data would have some insights into gene expression profile during spermatogonial differentiation and meiotic initiation in in vitro primary spermatogonial culture. However, overall manuscript is descriptive with short of supporting data. Major concerns in this manuscript are raised from some large gaps in the logics between the data and their assumption. The most serious problem is that what they supposed "the nutrient restriction" condition was not clearly defined. Because the assumption for the nutrient restriction" in the culture was not well defined, it is hard to tell whether "the nutrient restriction" was indeed critical for meiotic initiation, as has been described in nitrogen starved condition for meiosis induction in yeast. Thus, their main conclusion is highly speculative and the title is over-presentation. Therefore, the manuscript is hard to read and difficult to understand, raising several major concerns as described below.

(1) Page3 line line73-80, Fig1

The authors analyzed cell morphology and transcriptome using RA only, NR only, NRRA in Fig1 and following other data in later Figures. They claimed that NRRA treatment, but not RA or NR alone, induced meiotic DSBs and that distinct profiles of RNA-seq data were observed in different conditions. According to supplementary table S8, nutrient restriction medium contains 90% (for what?) + 10%SSC medium plus 10nM RA based on essentially same composition of SSC medium. This is confusing, because nutrient restriction medium already contained 10nM RA. Moreover, what stands for meiotic process medium? The authors should clarify the exact composition for RA and NR condition. Another concern is that nutrient restriction condition contained 1% FBS and 5%KSR as in other examined conditions, which indicates major nutrient was still in the medium rather than restricted. Further, even without addition of 10nM RA, FBS may still have contained undefined components. Thus, the authors cannot exclude a possibility that NR condition may have residual RA or retinol a metabolic

precursor of RA. The authors should show comparable data in the presence of RAR antagonist in NR condition.

From the manuscript, it is felt that the meiosis initiation and the cell differentiation were confused. However, these are independent events. Please discuss each event separately.

(2) Page3 line line87, FigS1D

They should show immunostaining images of cell under RA alone and NR alone.

(3) Fig1B, C do not support their conclusion that nutrient restriction was necessary and sufficient for induction of Spo11-dependent meiotic DSB program as described in summary. Thus, this should be rephrased.

(4)Page4 line line123, FigS7

They described that Gm4969 expression was augmented, but Gm4969 was missing in the heatmap in FigS7.

(5)They should describe clearly the data in Fig1H legend. For example, what is the difference between NR/RA-upregulated and NRRRA-upregulated?

(6) Page7 line line201, Fig3A

The authors analyzed DSB formation in Stra8-deficient culture. However, it is hard to see DMC1 foci in Fig3A. Please show more clear images and quantify the number of foci rather than % of DMC1 positive cells.

(7) Page8 line line234, Fig4D

The authors claimed that DSBs were detected at hotspot by DMC1-ChIP. They should show the exact location of the hotspot rather than simply indicating chromosome number.

(8) Page35 FigS22 legend

In FigS22 A, early pachytene-like cells were defined by RAD51, DMC,1 MEIOB and SPATA22 foci with SYCP3 immunostaining. However, those early pachytene-like cells look like earlier stage rather than pachytene-like because chromosome axes labeled by SYCP3 are discontinuous and patchy. They should clearly describe how staging was done.

(9) Discussion : Page9 line line257-275

The authors claimed that NR mimics the apical environment of seminiferous tubules, where they suppose that nutrient would be less rich. Please show the reference for such evidence. Their conclusion is that meiotic initiation is regulated by two independent pathways, RA and NR. However, is this idea applied to meiotic initiation in embryonic ovary? Please discuss on meiotic initiation in embryonic ovary. Whole through the manuscript, I cannot find any evidence and description that support their idea that inducer of yeast meiosis is critical for meiotic initiation in mammalian germ cells. What did the authors mean by "inducer of yeast meiosis"? More comprehensive descriptions are required.

(10) Page29 line: Materials and Methods: Statistical analysis

The authors described that data were analyzed by Student's t-test. However, this is not suitable for multiple groups. Please re-analyze the data using appropriate method for multiple groups such as Fig.1B.

(11) Discussion about relationship between RA and NR is critical to support the author's claim. For this purpose, it is required to analyze what kind of changes were induced in each signal separately and combinatory used condition in deep. Please describe this.

(12) Page3 and 4 line 89-104

The authors showed meiotic genes were significantly upregulated in NRRRA. The authors also claimed that NRRRA induce spermatogonial cell differentiation. Therefore, it is expected that stem cell maintenance related genes were downregulated in NRRRA condition, However, the authors have not mentioned about that point. The stem cell differentiation is also critical step to make haploid. Please analyze and discuss clearly about differentiation from stem cells in NRRRA treated condition.

Reviewer #3 (Remarks to the Author):

In this manuscript, the authors recapitulated early meiosis in vivo and tried to show that nutrient restriction is both necessary and sufficient to robustly induce Spo11- dependent meiotic DNA double strand breaks (DSBs) and Stra8-dependent meiotic gene programs with RA. They incorporated single-cell RNA-seq and bulk RNA-seq data to demonstrate their findings. Overall speaking, it is very interesting. However, some analysis steps and results are confusing. The following comments need to be addressed before it meets publication requirements.

SEP

Analysis of single-cell RNA-seq data

1. Authors listed the marker genes for the clusters on Page 5, line 138-144: cluster 0(e.g., Gfra1, Evt5), cluster 1-3 (e.g., Sohlh1, Dmc1, Sycp3), cluster 4(s100a4). However, they are inconsistent with the genes listed in Figure 2C: Gfra1, Cyp26a1, Ccnb1, Prdm9, Sycp1, Ddx4. Is there a specific reason for that?
2. Figure 2D, the current color palette cannot tell the difference between clusters 0 and 1 very well. Since they are less than five colors, maybe authors can find some other colors that can better distinguish clusters?
3. Figure 2E & J, I wonder why cluster 4 was not shown.
4. Figure 2G, a cluster of single cells was visualized as one point in the PC plot. It is unclear how they performed the PCA. For example, did the authors aggregate the cells as a pseudobulk here?
5. Authors used "relative levels" in some figures (e.g. S11) and "relative expression" in some others (e.g. S15). Are they referring to the same concept?
6. The authors wrote, "Genes expressed in less than 3 cells were filtered out and cells with expressed genes less than 200 were excluded." (page 29, line 1-2). Did the authors apply CreateSeuratObject() from Seurat package with the default settings (min.cells = 3, min.features = 200)? If yes, have the authors checked whether these settings are applicable based on the distribution of the current dataset? Otherwise, if authors implemented by themselves, according to the order mentioned, they filtered lowly expressed genes first and then filter low-quality cells, it could lead to some potential problems because genes were filtered using low-quality cells.
7. The authors wrote, "Seurat object was built after filtered and normalized with default settings." (Page 29, line 3). Does this imply filtering and normalization were performed before applying Seurat package for further processing data? It was not clear what normalization method was applied here (for example, logCPM).
8. The authors wrote, "cells with expressed genes less than 200 were excluded" (page 29, line 2-3), and "cells were retained only if they contained > 2000 ufeature" (page 29, line 4). These two claims seem contradictory. It is unclear which one has been applied.
9. It has not been mentioned what the word "ufeature" (page 29, line 4) is referring to. Since the authors applied Seurat, I suppose that is "nfeature". It is suggested to modify the texts as "number of genes" or "number of features" to adapt to a large group of readers who might not necessarily be familiar with Seurat.
10. Why top 6000 highly variable genes and how were they identified? Fig. S9E shows the standardized variance of many of these genes is around 0.
11. Have the authors compared tSNE with UMAP which was a more recently developed and widely used dimension reduction method? Will the current results be reproducible on UMAP?
12. The sentence "Cluster define was by specific marker genes using "FeaturePlot" function."(line 8) is

grammatically incorrect. Besides, FeaturePlot() is a function in the Seurat package to visualize gene expression across cells. It is unclear how a plot function defined clusters. FindClusters() can be used to identify clusters with specifying a resolution. The authors need to refer to a specific list of marker genes and the plots of the marker genes (Figure S10).

13. "Differential gene expression analysis for different cell types was by using the Monocle." It is not clear how they identified the "cell types" here.

14. The authors used Monocle to identify differentially expressed genes. Monocle can also be applied to construct pseudotime trajectories. However, the authors used slingshot instead to infer trajectories. Is there a specific reason for that? What would the result look like if the authors stick to Monocle?

15. There is only one lineage for all clusters of cells, shown in Figure S15D. Monocle identified differentially expressed genes across to pseudotime. It is confusing why these differential genes are marked for each cluster in Figure S16.

16. It has not been mentioned how the authors address the drop-outs in single-cell RNA-seq data which is an important issue in single-cell RNA-seq data analysis.

17. The authors provided a GitHub repository (<https://github.com/iamzhangxiaoyu/scRNA-seq>) for the source of codes, however, no codes can be seen there (July 25, 2020, 9:00AM).

18. The authors need to add substantial citations for all the packages and methods they applied and specify the versions of each of the packages. For example, Seurat, JackStraw procedure, tSNE, Monocle, Slingshot, Destiny, pheatmap, etc..

SEP:

Data Availability

1. Authors mentioned "FastQ files of single-cell RNA-seq and bulk RNA-seq are available on GEO GSE" (Page 29, last line) but there is no GEO number provided. Is there a specific reason for that?

Point-to-point response to reviewers' comments (NCOMMS-20-24265-T):

Reviewer #1:

In this manuscript, Zhang and colleagues use an *ex vivo* mouse primary spermatogonia culture system to examine factors associated with meiotic initiation in the male germline. The authors report that with dual treatment of Earle's Balanced Salt Solution ("starvation" media) and Retinoic Acid they observe hallmarks of meiotic entry, including formation of SPO11-induced DSBs, expression of meiosis-related genes, and cytological evidence of meiotic chromosomes. Importantly, the authors show genetically that formation of the DSBs are dependent on the presence of *Stra8* as well. Using single-cell RNA-seq, Zhang et al. also compare their culture system to published single-cell RNA-seq datasets of *in vivo* male germ cells and show that their cultures are comprised of cells with gene expression profiles similar to undifferentiated spermatogonia all the way through to pachytene-stage spermatocytes. Finally, the authors identify 11 candidate transcription factors which are upregulated in the nutrient-restricted condition (independent of RA) and hypothesize that these are factors required for meiotic initiation.

Overall, this manuscript performs many genomic analyses of cells produced in the culture system, and makes an effort to validate the *ex vivo* findings using relevant *in vivo* datasets and genetics. While it is apparent that NRRA is inducing meiotic initiation *ex vivo*, the evidence for the same thing occurring *in vivo* is not nearly as convincing. One important issue with the manuscript is that there are a number of places where there is unclear/insufficient rationale for experiments and interpretation of data. This paper was clearly drafted originally for a short report format, but I believe that this should be expanded into a full article format so as to clarify the presentation. In some cases, it is unclear to me whether problematic portions are the result of insufficient/unrigorous experimentation, or just brevity. Below are specific comments.

We appreciate that the reviewer considers our *in vitro* results convincing, which show that nutrient restriction in combination with retinoic acid (NRRA) induces meiotic initiation in primary culture of mouse spermatogonia (or SSC culture) that faithfully recapitulates the transcriptomic and cytologic features of *in vivo* meiosis, by commenting "it is apparent that NRRA is inducing meiotic initiation *ex vivo*". These results demonstrate a conserved role of nutrient restriction, inducer of meiosis in the yeast system, in inducing mammalian meiotic initiation.

Based on these results, as well as past studies, which show that blood-testis barrier (BTB) creates a nutrient-restricted microenvironment for meiotic spermatocytes, our study raises the possibility that nutrient-sensing pathway may play a role in regulating meiosis *in vivo*, which we discussed in the Discussion section of the manuscript (line 387 – 411). We did not claim "the same thing occurring *in vivo*" in our manuscript. To make this clear, we modified the title of our manuscript to "Nutrient restriction, inducer of yeast meiosis, induces mammalian meiotic initiation *in vitro*".

We'd like to emphasize that, to date, a signal that is sufficient to induce meiotic initiation in mammalian germ cells is not known. Since NRRA is the first defined signal sufficient to induce meiotic initiation without gonadal somatic cell support in culture, our study provides novel

insight into the molecular mechanism underlying mammalian meiotic initiation from the perspective of nutrient-sensing pathway. As an example, in our manuscript, we have uncovered a network of 11 transcription factor genes, whose expressions are upregulated by nutrient restriction but not RA. Intriguingly, 2 transcription factor genes (*Sohlh1*, *Sox3*) are known to play critical roles in meiosis and spermatogenesis. Moreover, these transcription factor genes are expressed during early meiosis *in vivo*, suggesting that nutrient-sensing pathway is implicated in this process. Moreover, given that meiotic induction *in vitro* still relies on gonadal somatic cell support, our study significantly advances the technique of *in vitro* gametogenesis.

Note to the reviewer: we conducted our experiments using long-term SSC culture but not freshly isolated spermatogonia. Thus, our experiment is under “*in vitro*”, but not “*ex vivo*”, setting.

We have now revised the manuscript to a full-length article, which contains 8 figures (60 panels), 25 supplementary figures, 10 supplementary tables. Of these data, 15 panels of new or revised data were added to this revised manuscript, based on reviewers’ constructive comments, to strengthen our conclusion that nutrient restriction is an addition signal to RA that induces meiotic initiation in mammalian germ cells and enhance the rigor of our study. Please refer to our revised manuscript for details.

Major comments:

1. The specific nature of the Earle’s Balanced Salt Solution-based nutrient restriction is unclear. What nutrients are being restricted when EBSS is used? Why was EBSS chosen as the nutrient restriction agent? Would any agent which caused nutrient restriction be expected to yield similar results?

We have tested the effect of Earle’s Balanced Salt Solution (EBSS), Hank’s Balanced Salt Solution (HBSS), and phosphate buffered saline (PBS), three commonly used buffers to starve cells. However, when cultured in HBSS- or PBS-containing nutrient restriction medium, cells were often detached from tissue culture plates (common to other cell types as well). Thus, we chose to use EBSS, but not HBSS or PBS, in all of our following experiments. This information is included in the revised manuscript, line 89 – 97.

2. The authors do not discuss the extensive literature on *in vitro* spermatogenesis that has emerged in recent years, including differentiation of pre-spermatogonial cells. It might be useful to compare the efficiencies and the extent to which germ cells develop in these other systems.

Discussions on *in vitro* spermatogenesis are now included in both the “Introduction” section of our revised manuscript, which focuses on *in vitro* spermatogenesis using spermatogonial stem cells (line 58 – 66), and the “Discussion” section of our revised manuscript. It is important to note that, the study to induce meiosis using either spermatogonial stem cells or pluripotent stem cell-derived cells is scarce. Our study is the first to show Spo11-dependent meiotic DSB formation using DMC1, RAD51, MEIOB, SPATA22, the first to quantify the number of meiotic DSBs, and the first to map the genomic distribution of these meiotic DSBs. Similar studies have not been performed before.

By reviewing the literature, it is clear that our study is the first to achieve robust meiotic initiation without gonadal somatic cell support. Additionally, by establishing *Spo11*-deficient and *Stra8*-deficient SSC culture, our study shows that meiotic DSBs and gene program induced by NRRA requires *Spo11* and *Stra8* (Figs. 1, 4, and 5). Moreover, we have conducted extensive transcriptomic analysis by using both bulk RNA-seq (Figs. 1 and 4) and scRNA-seq (Fig. 2) to characterize the transcriptome feature of *in vitro* meiotic initiation and compare it with *in vivo* meiosis. Furthermore, we have also quantified the number of meiotic DSBs formed *in vitro* and mapped whether they are distributed to a few well-characterized strong hotspots of homologous recombination (Fig. 6). Similar experiments have not been conducted in the context of *in vitro* spermatogenesis using SSC culture before.

3. Antibody staining in Fig. 1B-D should be done on meiotic chromosome preparations, not whole cells. If these cells are really undergoing meiosis, then the chromosomes should be assuming a true meiotic appearance. Later figures have chromosome spreads. Nevertheless, there are no examples of pachytene spermatocytes. Is there no progression to this stage with further culture?

We have conducted further staining using meiotic chromosome preparations. By using colocalization of SYCP1, a late component of central element of synaptonemal complex, and SYCP3, a lateral element of synaptonemal complex, we show that cultured spermatogonia undergo meiotic initiation and progression to a stage similar to early pachytene (Fig. 3). In addition, we show that meiotic progression *in vitro* to reach early pachytene stage requires *Stra8* and *Spo11* (Fig. 5).

While meiotic chromosome preparations are instructive in determining the meiotic stages, we would like to emphasize that detection of SYCP1 and SYCP3 co-localization in thread-like morphology in culture cells as shown in Figs. 3 and 5 also provides strong evidence that the cultured cells were advancing in meiotic prophase *in vitro*, and this effect requires both RA and nutrient restriction (Fig. 3D). To our knowledge, detection of SYCP1 and SYCP3 co-localization in thread-like morphology in culture cells has not been reported before.

4. Lines 106-111. Just because CD1 mice had similar results doesn't mean one can confidently say that "the sufficient role of NRRA in inducing meiotic gene program is independent of genetic backgrounds". Only two backgrounds were tested, the first being an F1 hybrid.

We have revised the manuscript as: "these data suggest that the effect of NRRA in inducing meiotic gene program does not depend on a specific genetic background" (line 138 – 139).

5. Clarification about the MEF feeder layer is important. I assume the primary MEFs were rendered mitotically inactive? If so, how was this done? ... The methods should clarify the mouse strain used to generate the MEFs.

MEFs were purchased from Thermo Fisher (cat# A34180). According to manufacture website, MEF cells are collected from outbred CF1 mice, mitotically arrested by irradiation.

This information has now been included in our manuscript. It is important to note that MEF cells were used in parallel for cultures under both control and treatment conditions.

6. Lines 132-133. Elaborate on "...a medium that is supposedly to promote meiotic progression for an additional 1 and 2 days."

We have provided additional details on the medium to promote meiotic progression in lines 164 – 168 of the revised manuscript, which we referred to as "meiotic progression" medium. In the revised Supplementary Table 8, the compositions for all media including their companies and catalog numbers are provided.

This "meiotic progression" medium is formulated by omitting GDNF and FGF2, cytokines that maintain stem cell renewal, from the complete SSC medium to allow for spermatogonial differentiation. To support meiotic progression, we included melatonin (10^{-7} M), follicle-stimulating hormone (FSH) (200 ng/ml), transforming growth factor (TGF)- β (10 ng/ml), bovine pituitary extract (BPE) (50 ng/ml), and dihydrotestosterone (DHT) (10^{-6} M).

7. Fig. S22 purports to show early pachytene-like spermatocytes. That is clearly not the case; they look like late leptotene or maybe zygotene at best. It would be a good idea to comment at some point on the timing of progression from pre-pubertal gonial to putative pachytene spermatocytes in this system compared to *in vivo*. It seems like 2-4 days as reported here is extremely rapid.

We have now generated new data presented in **Fig. 3** to examine whether NRRA-induced activation of meiotic gene program supports meiotic progression in prophase based on synaptonemal complex formation. In **Fig. 3**, we conducted co-localization staining for cultured cells in both culture dish and their chromosome spreads using SYCP3, a lateral element of synaptonemal complex, and SYCP1, a late component of central element of synaptonemal complex. While nuclear accumulation of SYCP3 appeared between day 1 – 2 (pre-leptotene-like stage) when cultured cells were treated with NRRA medium, intermittent SYCP3 staining began to appear in chromosomal axis from day 3 when cells were switched to the "meiotic progression" medium, suggesting a leptotene-like stage. Between day 3 – 4, SYCP1 foci were detected, suggesting an early zygotene-like stage. Between day 4 – 5, gradual colocalization of SYCP3 and SYCP1 appeared, suggesting a late zygotene-like stage. Between day 5 – 6, extensive colocalization of SYCP3 and SYCP1 in a thread-like morphology were observed, suggesting that cultured cells had reached to an early pachytene-like stage.

Based on these observations, it takes approximately 5 to 6 days for cultured cells to reach early pachytene stage. Under *in vivo*, first wave of meiosis begins from postnatal day 7 – 8 and pachytene spermatocytes are observed in testes at postnatal day 15. Thus, kinetics of meiotic progression is comparable under both *in vitro* and *in vivo* conditions. This information has been added to our revised manuscript (line 250 – 252).

8. Lines 231-235. No convincing data, or insufficient data, is shown to support the conclusion that the NRRA-induced meiosis has the same recombination hotspots as *in vivo*. Fig. S22b

does not match the legend, which says that two cell types were compared. The exact locus coordinates were not given. This dataset is not acceptable for publication as presented or discussed.

We apologize for the error in the legend, which has now been corrected. We have now provided the exact locus for recombination characterized in **Fig. 6D**. Although we believe the genomic distribution of meiotic DSBs for recombination is an essential criterion to evaluate how faithful *in vitro* meiosis recapitulates *in vivo* meiosis, this experiment has not been conducted before. Thus, our study presents the first effort in literature to characterize recombination hotspot for *in vitro* meiosis.

9. Along the lines of MEFs, Cluster 4 (comprising of the MEFs) is absent in Figure 2D+E. I think readers would be interested in seeing Cluster 4 represented in those figures.

Based on high level of *S100a4* gene expression (**fig. S10D**), Cluster 4 represents irradiated MEF cells that were used as feeder cells that support the growth of primary spermatogonia during *in vitro* culture. We generated the tSNE plot, which pooled samples from different timepoint. The plot (next page) shows the proportions of cells in Cluster 4 are consistent at each timepoint. Since our study is focused on characterizing the meiotic gene program in germ cells, which express *Ddx4* in Clusters 0, 1, 2, and 3, Cluster 4 (irradiated MEF cells) are not included in the Figures 2D and E.

10. The last sentence of the paper mentions that the *in vitro* platform can facilitate both the study of meiotic initiation and *the production of haploid gametes in culture*. What happens to the cells after the completion the treatments described? Do they continue onwards in the meiotic program?

Meiotic initiation is the focus of our study, and we have not tested culturing the cells for an extended period. By overcoming the barrier of meiotic gene activation that supports meiotic DSBs and synapsis, we consider that our study may ultimately facilitate production of haploid gametes via meiosis.

11. Lines 67-68: “We established primary spermatogonial culture by using neonatal mouse testicular cells in C57BL/6 X DBA/2 F1 hybrid background.” These were from whole testes, but it is mentioned in the methods that “stem cell enrichment” was performed. What is that? How enriched? Is it possible that there are testis somatic cells in the cultures, in addition to the MEF feeders? This needs to be validated in detail.

Stem cell enrichment was done by differential plating (Ref. 1 in Materials and Methods). We have now provided more details about the procedures. To establish SSC culture, testes from 6-day-old mouse pups were dissociated by a two-step enzymatic digestion procedure. Briefly, after removal of the tunica albuginea, testes were cut into small pieces and transferred to a 15-ml tube containing 4 ml of collagenase IV solution (1 mg/ml in HBSS). Tissues were incubated in 37°C water bath for 20 minutes when tissues are separated. After washing the tubule fragments with 4 ml of HBSS and centrifuging at 300g for 5 minutes, the tubule fragments were incubated in 4 ml of 0.05% trypsin-EDTA solution in 37 °C water bath for 5 min. Testicular cells were cultured on 0.1% gelatin-coated plates with complete SSC medium overnight. Then, non-adherent cells (germ cells) were collected, while adherent cells (somatic cells, e.g., Sertoli cells) were discarded (differential plating to enrich stem cells). Collected non-adherent cells (germ cells) were transfer on to MEF feeder cells for expansion and long-term culture.

Under this protocol, testicular somatic cells were not included in SSC culture. This is confirmed by scRNA-seq, which did not detect a cell cluster expressing Sertoli cell marker genes (e.g., *Sox9*), and staining of SSC culture using *Sox9* (see picture below).

12. Last paragraph of text. The authors claim that their results are consistent with the *in vivo* situation, in which the BTB causes a nutrient starved domain that triggers meiosis. However, the first wave of spermatogenesis in mice occurs before the BTB is formed, beginning 15

days postpartum. This begs the question of whether the results here, which involve neonatal germ cells, does not accurately represent the situation of meiotic entry in adults

We agree with the reviewer that complete closure of BTB may begin around 15 days postpartum. However, we'd like to point out that expression of some BTB components (e.g., Cx43, a gap junction protein,) commences in mouse testes from ~6 days of age, concomitant with the first wave of meiotic initiation. Interestingly, *Cx43* is required for meiotic initiation, in that conditional deletion of *Cx43* in Sertoli cells results in meiotic arrest at spermatogonia stage. Thus, we consider that individual BTB protein components play distinct roles in controlling substance flow across the BTB and that some BTB component(s), e.g., Cx43, may have a specific role in regulating meiotic initiation. This new information has now been included in the "Discuss" section of our revised manuscript (line 387 – 411).

Our current manuscript presents our findings that nutrient restriction, inducer of yeast meiosis, in combination of RA is sufficient to induce meiotic initiation in SSC culture. For the first time, meiotic initiation is robustly induced in cultured germ cells in the absence of gonadal somatic cell support. In addition to providing a practical and easily reproducible approach to induce meiosis in cultured germ cells, our study provides a new dimension to dissect the molecular mechanism underlying meiotic initiation in mammalian germ cells through nutrient-sensing pathway.

Minor comments:

1. Line 56: The mesonephric ducts are not part of the ovary.

We have revised the manuscript as: "In female oogenesis, RA is synthesized primarily in the mesonephric ducts to which the embryonic ovaries are attached" (line 48 – 49 in the revised manuscript).

2. Page 4, line 116: "Unsupervised UHC" is redundant

The redundant word, "unsupervised", has been deleted. Please refer to line 145 in the revised manuscript.

3. Page 5, line 132: "supposedly" should be "supposed"

We have corrected the word to "supposed". Please refer to line 162 in the revised manuscript.

4. Page 8, line 223: "sing-strand" should be "single-strand"

We apologize for the typo. The text has been corrected to "single-strand". Please refer to lines 314 in the revised manuscript.

5. Lines 15-17 of the Abstract are confusing: “nutrient restriction is both necessary and sufficient to robustly induce Spo11-dependent meiotic DNA DSBs and Stra8-dependent meiotic gene programs” but then continues to say “with RA...” Is NR necessary and sufficient for meiotic induction or is RA also needed? If RA is also needed, then it doesn't seem like NR is sufficient for meiotic induction...

Both nutrient restriction and RA are required for meiotic initiation. The text has been revised to: “we show that nutrient restriction in combination with RA is sufficient to robustly induce *Stra8*- and *Spo11*-dependent meiotic gene and chromosome programs that faithfully recapitulate the transcriptomic and cytologic features of *in vivo* meiosis”. Please refer to lines 14 to 17 in the revised manuscript.

6. Page 8, lines 243-245: This sentence describing a result is also very confusing.; “Using *Dmc1*, *Sycp3*, *Hormad1* as three representatives, we found that 11 of them...”

The text has been revised to “using *Dmc1*, *Sycp3*, *Hormad1* as representative meiotic genes, we found that 11 nutrient restriction-upregulated TF genes them shows strong (Pearson correlation > 0.6) or moderate correlation (Pearson correlation > 0.4) with meiotic gene expression”. Please refer to lines 334 to 337 in the revised manuscript.

7. It might be worthwhile to comment on how the timing of your NRRA treatment conditions compares to the timing/number of days it takes for meiotic initiation *in vivo*. Is the timing comparable? Or is *ex vivo* initiation faster?

Please refer to our response to major comment #7. “Based on these observations, it takes approximately 5 to 6 days for cultured cells to reach early pachytene stage. Under *in vivo*, first wave of meiosis begins from postnatal day 7 – 8 and pachytene spermatocytes are observed in testes at postnatal day 15. Thus, kinetics of meiotic progression is comparable under both *in vitro* and *in vivo* conditions.” This information has been added to our revised manuscript (line 250 – 252).

8. Page 7, lines 208-212: This sentence related to the interpretation of the data as currently written is very confusing and must be fixed: “...in contrast to WT and Spo11-deficient cultures, despite activation of retinoid acid receptor signaling pathway, gametogenesis- and meiosis-related pathways were not activated in *Stra8*-deficient culture upon NRRA treatment.”

We have revised this part of the text to:

GSVA analysis shows that STRA8 sits at the foundation of NRRA-induced meiotic gene program *in vitro*, in that gametogenesis- and meiosis-related pathways were not activated in *Stra8*-deficient culture upon NRRA treatment (**Fig. 4C**). This is consistent with the essential role of STRA8 in activating gene program of meiotic prophase^{2, 14, 34}, and confirms that lack of meiotic DSBs in *Stra8*-deficient and *Spo11*-deficient cultures resulted from discrete mechanisms. In *Spo11*-deficient culture, despite normal activation of meiotic gene program, meiotic DSBs were not formed due to absence of Spo11 (**Fig. 1D**), the enzyme directly

responsible for this process^{4,35}. In *Stra8*-deficient culture, genes with GO term “retinoid acid receptor signaling pathway” was activated, suggesting that the failure *Stra8*-deficient culture to activate meiotic gene program did not result from a lack of RA signaling response (Fig. 4C).

Please refer to lines 278 to 288 in the revised manuscript.

9. I do not believe there was any mention of what the percentages below the immunofluorescent images indicate. I am assuming it means the percentage of cells seen with the relevant staining under different conditions (?), but it should be stated in figure legends/where appropriate.

The percentages below the immunofluorescent images indicate the percentage of cells seen with the relevant staining under different conditions. This information has been added to the figure legends for Figs. 1C and D and Fig. 4A in the revised manuscript.

10. Like Supplemental Table S9, Table S8 should include the company, catalog and lot numbers associated with the media components. This is important because tissue culture media components can be highly variable between companies and lot numbers.

We have now revised Table S8 to include the companies and catalog numbers associated with all medium components. Our experiments are highly reproducible and do not exhibit lot number differences.

11. In acknowledgements, it says that Spo11 mutant mice were obtained from Jasin/Keeney, but in the Methods, it says they were obtained from JAX. This (and several other poor descriptions throughout the paper) concerns me in general about the rigor of the research.

We apologize for the confusion. Spo11-mutant mice were obtained from Jasin/Keeney. This mouse strain is also available from Jax (stock number: 019117). This information has now been included in the Methods. In addition, we have revised the entire manuscript to a full-length article as the reviewer kindly suggested in order to describe additional details wherever possible.

Reviewer #2:

The authors investigated the signals to initiate meiosis in mouse. They performed scRNA-seq analyses using primary spermatogonial culture to see how different culture conditions affected on transcriptome change for spermatogonial differentiation and activation of meiotic genes. They found that transcriptome in primary spermatogonial cells was different under nutrient restriction (NR), retinoic acid (RA) exposure, and combination of both NR and RA compared to untreated control. The authors claimed that nutrient restriction was necessary and sufficient for induction of Spo11-dependent meiotic DSB program and *Stra8*-dependent meiotic gene program, which led them to the conclusion that nutrient restriction is the conserved mechanism for meiotic initiation as observed in yeast. Their scRNA-seq data would have some insights into gene expression profile during spermatogonial differentiation and meiotic initiation in in vitro primary

spermatogonial culture. However, overall manuscript is descriptive with short of supporting data. Major concerns in this manuscript are raised from some large gaps in the logics between the data and their assumption. The most serious problem is that what they supposed “the nutrient restriction” condition was not clearly defined. Because the assumption for the nutrient restriction” in the culture was not well defined, it is hard to tell whether “the nutrient restriction” was indeed critical for meiotic initiation, as has been described in nitrogen starved condition for meiosis induction in yeast. Thus, their main conclusion is highly speculative and the title is over-presentation. Therefore, the manuscript is hard to read and difficult to understand, raising several major concerns as described below.

Based on comment #1 below, a major reason that causes the confusion of reviewer 2 is because we made a typo in our Table S8, which inadvertently listed retinoic acid in the composition of our nutrient restriction medium. Please note that our nutrient restriction medium does not contain retinoic acid. We apologize for this error, which has now been corrected. We have improved the clarity of Table S8 to show the compositions of each medium used in our study. As suggested by the reviewer, to down-tune the title of our manuscript, we have revised it to “Nutrient restriction, inducer of yeast meiosis, induces mammalian meiotic initiation *in vitro*”.

1. Page 3, line 73 – 80, Fig. 1. The authors analyzed cell morphology and transcriptome using RA only, NR only, NRRA in Fig. 1 and following other data in later Figures. They claimed that NRRA treatment, but not RA or NR alone, induced meiotic DSBs and that distinct profiles of RNA-seq data were observed in different conditions:
 - a. According to supplementary table S8, nutrient restriction medium contains 90% (for what?) + 10% SSC medium plus 10 nM RA based on essentially same composition of SSC medium. This is confusing, because nutrient restriction medium already contained 10nM RA.

We again apologize for the confusion caused by our typo in the Table S8: nutrient restriction medium does not contain retinoic acid. We have improved the clarity of Table S8 to show the compositions of each medium used in our study. As shown by our extensive data (Figs. 1 and 3), nutrient restriction medium alone does not induce meiotic initiation; it is nutrient restriction medium in combination of RA that is sufficient to induce meiotic initiation *in vitro*.

- b. Moreover, what stands for meiotic process medium? The authors should clarify the exact composition for RA and NR condition.

This “meiotic progression” medium is formulated by omitting GDNF and FGF2, cytokines that maintain stem cell renewal, from the complete SSC medium to allow for spermatogonial differentiation. To support meiotic progression, we included melatonin (10^{-7} M), follicle-stimulating hormone (FSH) (200 ng/ml), transforming growth factor (TGF)- β (10 ng/ml), bovine pituitary extract (BPE) (50 ng/ml), and dihydrotestosterone (DHT) (10^{-6} M).

It is important to note that although this “meiotic progression” medium promotes meiotic progression to early pachytene stage, it alone does not induce meiosis (**Fig. 3D and E**).

- c. Another concern is that nutrient restriction condition contained 1% FBS and 5%KSR as in other examined conditions, which indicates major nutrient was still in the medium rather than restricted.

Complete SSC medium contains 1% FBS and 5% KSR. Nutrient restriction medium contains 10% SSC medium and 90% EBSS. Thus, in the nutrient restriction medium, only 0.1% FBS and 0.5% KSR are present.

A direct cellular response to nutrient restriction is autophagy. We show that treatment of SSC culture to this nutrient restriction medium is sufficient to induce autophagy activation, suggesting that this nutrient restriction medium induces nutrient restriction to cells (**fig. S1B**). This information has been added to our revised manuscript (lines 95 to 97).

- d. Further, even without addition of 10nM RA, FBS may still have contained undefined components. Thus, the authors cannot exclude a possibility that NR condition may have residual RA or retinol a metabolic precursor of RA. The authors should show comparable data in the presence of RAR antagonist in NR condition.

Nutrient restriction (NR) alone condition does not induce meiotic initiation (**Figs. 1 and 3**). Thus, even if there is residual RA or retinol, we consider that it is not necessary to test RAR antagonist in NR condition.

- e. From the manuscript, it is felt that the meiosis initiation and the cell differentiation were confused. However, these are independent events. Please discuss each event separately.

We agree with the reviewer that meiotic initiation and cellular differentiation are independent events. Thus, we have added a new figure to discern the differential regulation of cellular differentiation by retinoic acid (RA), nutrient restriction (NR), and NR plus RA (NRR) (**fig. 1E**). We found that that RA alone downregulates PLZF expression, a marker for undifferentiated spermatogonia, which is consistent with the role of RA in promoting cellular differentiation. However, NR shows no effect on PLZF expression, suggesting a specific role for NR in synergizing with RA to induce meiotic initiation. This information has been included in lines 110 – 114 in the revised manuscript.

2. Page 3 line 87, Fig. S1D. They should show immunostaining images of cell under RA alone and NR alone.

In the revised figure (now fig. S1E), we show immunostaining of cells under all conditions.

3. Fig. 1B and C do not support their conclusion that nutrient restriction was necessary and sufficient for induction of Spo11-dependent meiotic DSB program as described in summary. Thus, this should be rephrased.

This sentence has been rephrased as the following (lines 15 to 18):

“we show that nutrient restriction in combination with RA is sufficient to robustly induce *Stra8*- and *Spo11*-dependent meiotic gene and chromosome programs that faithfully recapitulate the transcriptomic and cytologic features of in vivo meiosis. Neither nutrient restriction nor RA alone exerts these effects.”

4. Page 4 line 123, FigS7. They described that Gm4969 expression was augmented, but Gm4969 was missing in the heatmap in FigS7.

We have revised Fig. S7 to include *Gm4969* expression in the heatmap.

5. They should describe clearly the data in Fig. 1H legend. For example, what is the difference between NR/RA-upregulated and NRRA-upregulated?

“NR/RA-upregulated” indicates genes that were upregulated by NR or RA. “NRRA-upregulated” indicates that were upregulated by NRRA (not upregulated by NR or RA alone). We have included this information in the revised legend for **Fig. 1H**.

6. Page 7 line 201, Fig. 3A. The authors analyzed DSB formation in *Stra8*-deficient culture. However, it is hard to see DMC1 foci in Fig. 3A. Please show more clear images and quantify the number of foci rather than % of DMC1 positive cells.

We have conducted chromosome spread staining for DMC1 and quantified the foci number. Please see revised **Fig. 6**.

7. Page 8 line 234, Fig. 4D. The authors claimed that DSBs were detected at hotspot by DMC1-ChIP. They should show the exact location of the hotspot rather than simply indicating chromosome number.

Exact locations of the hotspots are now induced in **Fig. 6D**.

8. Page 35 Fig. S22 legend. In Fig. S22 A, early pachytene-like cells were defined by RAD51, DMC1, MEIOB and SPATA22 foci with SYCP3 immunostaining. However, those early pachytene-like cells look like earlier stage rather than pachytene-like because chromosome axes labeled by SYCP3 are discontinuous and patchy. They should clearly describe how staging was done.

To improve analysis of the stages of meiotic progression, we have now generated new data presented in **Figs. 3 and 5** based on synaptonemal complex formation. In **Figs. 3 and 5**, we conducted co-localization staining for cultured cells in both tissue culture dish and their chromosome spreads using SYCP3, a lateral element of synaptonemal complex, and

SYCP1, a late component of central element of synaptonemal complex. While nuclear accumulation of SYCP3 appeared between day 1 – 2 (pre-leptotene-like stage) when cultured cells were treated with NRRA medium, intermittent SYCP3 staining began to appear in chromosomal axis from day 3 when cells were switched to the “meiotic progression” medium, suggesting a leptotene-like stage. Between day 3 – 4, SYCP1 foci were detected, suggesting an early zygotene-like stage. Between day 4 – 5, gradual colocalization of SYCP3 and SYCP1 appeared, suggesting a late zygotene-like stage. Between day 5 – 6, extensive colocalization of SYCP3 and SYCP1 in a thread-like morphology were observed, suggesting that cultured cells had reached to an early pachytene-like stage.

9. Page 9 lines 257-275. The authors claimed that NR mimics the apical environment of seminiferous tubules, where they suppose that nutrient would be less rich. Please show the reference for such evidence. Their conclusion is that meiotic initiation is regulated by two independent pathways, RA and NR. However, is this idea applied to meiotic initiation in embryonic ovary? Please discuss on meiotic initiation in embryonic ovary. Whole through the manuscript, I cannot find any evidence and description that support their idea that inducer of yeast meiosis is critical for meiotic initiation in mammalian germ cells. What did the authors mean by “inducer of yeast meiosis”? More comprehensive descriptions are required.

We have revised the Discussions section of our manuscript to include the following requested information:

Extensive past studies have characterized the fluid composition inside seminiferous tubules and showed that it is low in glucose and most amino acids compared to blood plasma. These studies are nicely reviewed by Waites and Gladwell in a review article titled “Physiological significance of fluid secretion in the testis and blood-testis barrier” published in *Physiol Rev* 62, 624-671 (1982).

We speculate that the germ cell nest formed in fetal ovaries may provide a similar nutrient restriction function to induce meiotic initiation in primordial germ cells during oogenesis. Of note, one of the major protein components of blood-testis barrier required for meiotic initiation in spermatogenesis, connexin 43 or Cx43, is extensively expressed during fetal oogenesis. This information has been added to the Discussion section of our revised manuscript (line 387 – 411). Of note, our past analysis of *Stra8*-deficient mice revealed that a robust autophagy-inducing factor also acts on primordial germ cells in fetal ovaries.

“Inducer of yeast meiosis” refers to nutrient restriction, in that starvation induces yeast cells to transition from mitotic cell cycles to meiotic cell cycles.

10. Page 29 line: Materials and Methods: Statistical analysis. The authors described that data were analyzed by Student’s t-test. However, this is not suitable for multiple groups. Please re-analyze the data using appropriate method for multiple groups such as Fig. 1B.

We have analyzed Fig. 1B using two-way ANOVA test, which shows that only NRRA, but not RA or NR alone, significantly induced *Spo11*, *Dmc1*, and *Sycp3* expressions.

11. Discussion about relationship between RA and NR is critical to support the author's claim. For this purpose, it is required to analyze what kind of changes were induced in each signal separately and combinatory used condition in deep. Please describe this.

To discern the effect of RA, NR, and NRRA in meiotic gene program, we have analyzed the effect of RA, NR, NRRA by using RNA-seq analysis as shown in Fig. 1D. We assembled a set of 193 meiotic genes by combining two collections of genes that were previously found to be associated with early meiosis. Unsupervised hierarchical clustering (UHC) divided 165 differentially expressed genes (DEGs) into five major clusters (**Fig. 1E to H**). From this analysis, RA appears to activate a few essential meiotic genes, including *Stra8*, *Rec8*. NR appears to play a role in the activation of four classes of key meiotic genes by: 1) inducing expression of a subset of meiotic genes that were not regulated by RA, such as *Rad21l*; 2) inducing the expression of a subset of meiotic genes, which was further enhanced by RA, such as *Zfp541*; 3) synergizing with RA to induce the expression of a subset of meiotic genes, such as *Sycp1*, *Sycp3*, *Mei1*, *Msh5*, and *Stag3*; and 4) augmenting expression of a subset of meiotic genes induced by RA, including *Dmc1*, *Smc1b*, *Ugt8a*, *Stag3*, *Hormad1*, as well as *Gm4969*, which encodes MEIOSIN, a transcriptional cofactor for STRA8 required for meiotic prophase program. Thus, only in the presence of both NR and RA, meiotic gene program is robustly activated.

12. Page 3 and 4 line 89-104. The authors showed meiotic genes were significantly upregulated in NRRA. The authors also claimed that NRRA induce spermatogonial cell differentiation. Therefore, it is expected that stem cell maintenance related genes were downregulated in NRRA condition, However, the authors have not mentioned about that point. The stem cell differentiation is also critical step to make haploid. Please analyze and discuss clearly about differentiation from stem cells in NRRA treated condition.

We have added a new **fig. 1E** to discern the differential regulation of cellular differentiation by retinoic acid (RA), nutrient restriction (NR), and NR plus RA (NRRA). We found that while RA alone downregulates PLZF expression, a marker for undifferentiated spermatogonia, the addition of NR to RA further potentiates RA-induced cellular differentiation. NR alone has no effect on PLZF expression, suggesting a specific role for NR in synergizing with RA to induce meiotic initiation. This information has been included in lines 110 – 114 in the revised manuscript. In addition, in **fig. S11**, we show that marker genes for undifferentiated spermatogonia, namely *Gfra1*, *Zbtb16* (encoding PLZF), *Id4*, *Pouf51*, *Nanos2*, are rapidly downregulated by NRRA treatment, while meiosis-related genes are upregulated by NRRA.

Reviewer #3:

In this manuscript, the authors recapitulated early meiosis in vivo and tried to show that nutrient restriction is both necessary and sufficient to robustly induce Spo11-dependent meiotic DNA double strand breaks (DSBs) and Stra8-dependent meiotic gene programs with RA. They

incorporated single-cell RNA-seq and bulk RNA-seq data to demonstrate their findings. Overall speaking, it is very interesting. However, some analysis steps and results are confusing. The following comments need to be addressed before it meets publication requirements.

We thank the reviewer for considering our study “very interesting”. All comments raised by the reviewer regarding data analysis are addressed below.

Analysis of single-cell RNA-seq data

1. Authors listed the marker genes for the clusters on Page 5, line 138-144: cluster 0 (e.g., *Gfra1*, *Evt5*), cluster 1-3 (e.g., *Sohlh1*, *Dmc1*, *Sycp3*), cluster 4 (*s100a4*). However, they are inconsistent with the genes listed in Figure 2C: *Gfra1*, *Cyp26a1*, *Ccnb1*, *Prdm9*, *Sycp1*, *Ddx4*. Is there a specific reason for that?

To be consistent, we have updated the text as the following (lines 177 – 183):

Cluster 0 appears to be undifferentiated spermatogonia (e.g., *Gfra1*, *Evt5*). Clusters 1, 2, and 3 appear to be differentiating spermatogonia/pre- and meiotic spermatocytes at progressively advanced meiotic stages with upregulated expression of spermatogonial differentiation (e.g., *Ccnb1*, *Cyb26a1*, *Sohlh1*) and meiotic genes (e.g., *Sycp1*, *Prdm9*). Cluster 4 appears to be mostly somatic feeder cells (mouse embryonic fibroblast or MEF) due to the presence of fibroblast marker gene expression (*s100a4*) and absence of germ cell marker gene (*Ddx4*) expression.

Please note that *Gfra1*, *Ccnb1*, *Cyp26a1*, *Sycp1*, *Prdm9*, and *Ddx4* are presented in Fig. 2, *Evt5*, *Sohlh1*, and *S100a4* are presented in fig. S10.

2. Figure 2D, the current color palette cannot tell the difference between clusters 0 and 1 very well. Since they are less than five colors, maybe authors can find some other colors that can better distinguish clusters?

We have switched the color of cluster 1 to better discriminate clusters 0 and 1.

3. Figure 2E & J, I wonder why cluster 4 was not shown.

Based on its marker gene expression (*s100a4*), cluster 4 represents irradiated MEF cells that were used feeder cells during the experiments. They are no longer under physiological conditions (arrested by irradiation) and are not relevant to our analysis on meiotic initiation in *Ddx4+* germ cells. Thus, cluster 4 was not included in these data.

4. Figure 2G, a cluster of single cells was visualized as one point in the PC plot. It is unclear how they performed the PCA. For example, did the authors aggregate the cells as a pseudobulk here?

PCA was performed by merging our scRNA-seq data on *in vitro* meiosis with a published scRNA-seq data on *in vivo* meiosis. Then, we calculated the mean expression of genes in

each cluster by using monocle2 as show in github. The ‘ComBat’ function was used from the R/Bioconductor sva package to regress-out the technical effect. The first two PCs of the corrected expression were calculated by using the ‘prcomp’ R function. The information has now been included in the revised Materials and Methods section.

5. Authors used “relative levels” in some figures (e.g. S11) and “relative expression” in some others (e.g. S15). Are they referring to the same concept?

In fig. S11, “relative levels” indicate target gene expressions were normalized to the expression levels of a housekeeping gene, *Gapdh*, in each sample during quantitative PCR analysis. In fig. S15, which presents the analysis of a scRNA-seq dataset, we used a global-scaling normalization method "LogNormalize" that normalizes the feature expression measurements for each cell by the total expression, multiplies this by a scale factor (10,000 by default), and log-transforms the result.

6. The authors wrote, “Genes expressed in less than 3 cells were filtered out and cells with expressed genes less than 200 were excluded.” (page 29, line 1-2). Did the authors apply CreateSeuratObject() from Seurat package with the default settings (min.cells = 3, min.features = 200)? If yes, have the authors checked whether these settings are applicable based on the distribution of the current dataset? Otherwise, if authors implemented by themselves, according to the order mentioned, they filtered lowly expressed genes first and then filter low-quality cells, it could lead to some potential problems because genes were filtered using low-quality cells.

The CreateSeuratObject (... , min.cells = 3, min.features = 200) was used to create the Seurat object, resulting in a Seurat object with cells expressing at least 200 features, where each feature is expressed in at least 3 cells. We were satisfied with the gene expression distribution of this Seurat object, which ensures that known/expected genes are well represented across the cells and none are left out of the analysis. Additionally, we have also checked the distribution of the current dataset. As shown below:

```
## An object of class Seurat
## 31053 features across 25607 samples within 1 assay
## Active assay: RNA (31053 features, 0 variable features)
fivenum(apply(aggr@assays$RNA@counts,1,function(x) sum(x>0) ))
##      Gm1992      Ces2h Arhgap27os1      Zfp36      Gm42418
##           0           2           123           4190           25119
boxplot(apply(aggr@assays$RNA@counts,1,function(x) sum(x>0) ))
```

```
fivenum(apply(aggr@assays$RNA@counts, 2, function(x) sum(x>0) ))
## CCCGAAGCAGAATCGG-4 ATGACCAGTATCGGTT-4 CTTCTGTGGATGAC-3 TGCTTGCTCTAGTGTG-4
##                20.0                1217.5                4053.0                5231.5
## CATCCACGTTGCACGC-2
##                10098.0
hist(apply(aggr@assays$RNA@counts, 2, function(x) sum(x>0) ))
```

Histogram of `apply(aggr@assays$RNA@counts, 2, function(x) sum(x > 0))`

- The authors wrote, “Seurat object was built after filtered and normalized with default settings.” (Page 29, line 3). Does this imply filtering and normalization were performed before applying Seurat package for further processing data? It was not clear what normalization method was applied here (for example, logCPM).

We used `NormalizeData` (`normalization.method = "LogNormalize"`, `scale.factor = 10000`) from Seurat package to conduct normalization. The appropriate information has now been included in the revised Materials and Methods section.

8. The authors wrote, “cells with expressed genes less than 200 were excluded” (page 29, line 2-3), and “cells were retained only if they contained > 2000 ufeature” (page 29, line 4). These two claims seem contradictory. It is unclear which one has been applied.

We apologize for the confusion. We used ‘Subset’ function with (subset = nFeature_RNA > 2000 & percent.mt < 10), base on VlnPlot (features=c("nFeature_RNA", "nCount_RNA", "percent.mt"))s it shows in fig. S9. The appropriate information has now been included in the revised Materials and Methods section.

9. It has not been mentioned what the word “ufeature” (page 29, line 4) is referring to. Since the authors applied Seurat, I suppose that is “nfeature”. It is suggested to modify the texts as “number of genes” or “number of features” to adapt to a large group of readers who might not necessarily be familiar with Seurat.

We thank the reviewer for this kind suggestion. We have updated the word “ufeature” to “number of genes” in our revised manuscript.

10. Why top 6000 highly variable genes and how were they identified? Fig. S9E shows the standardized variance of many of these genes is around 0.

The genes selected to be used in this calculation has a major impact on the behavior of the metric and the performance of downstream methods. We want to select genes that contain useful information about the biology of the system, while removing genes that contain random noise. In the beginning, we used the default 2,000 highly variable genes for downstream analysis in Seurat. However, we found there was no effect on clusters compared with using 6,000 highly variable genes. Thus, to preserve more interesting biological structures, we used 6,000, instead of 2,000, highly variable genes during data analysis to improve computational reality of later steps.

11. Have the authors compared tSNE with UMAP which was a more recently developed and widely used dimension reduction method? Will the current results be reproducible on UMAP?

Yes, despite some slight differences in their ability to capture different features of the data, we have found that tSNE and UMAP produce qualitatively similar results. Notably, tSNE captures the time-dependent variation within each cluster for the time-course analysis better. Thus, in this manuscript we adopted tSNE throughout for visualization of single-cell expression profiles.

12. The sentence “Cluster define was by specific marker genes using “FeaturePlot” function” (line 8) is grammatically incorrect. Besides, FeaturePlot() is a function in the Seurat package to visualize gene expression across cells. It is unclear how a plot function defined clusters. FindClusters() can be used to identify clusters with specifying a resolution. The authors need to refer to a specific list of marker genes and the plots of the marker genes (Figure S10).

We apologize for the grammar error. We have corrected the sentence to “Clusters were defined by specific markers by using ‘FeaturePlot’ function”. The plots of the marker genes are showed in Fig. 2E (top 20 marker genes).

13. “Differential gene expression analysis for different cell types was by using the Monocle.” It is not clear how they identified the “cell types” here.

As differential gene expression analysis is built by using the Monocle in our study, we have transferred the cluster identified in Seurat to cell types in Monocle. The code used is shown in github.

14. The authors used Monocle to identify differentially expressed genes. Monocle can also be applied to construct pseudotime trajectories. However, the authors used slingshot instead to infer trajectories. Is there a specific reason for that? What would the result look like if the authors stick to Monocle?

In general, we have found that both Monocle and Slingshot produce qualitatively similar results (see below for the comparison), though with some slight differences. Most notably, for pseudotime trajectory analysis, Slingshot visualizes the pseudotime variation within each

cluster better. Thus, in our manuscript, we have adopted Slingshot throughout for visualization of single cell pseudotime trajectories.

15. There is only one lineage for all clusters of cells, shown in Figure S15D. Monocle identified differentially expressed genes across to pseudotime. It is confusing why these differential genes are marked for each cluster in Figure S16.

Different datasets are presented in figs. S15 and S16: fig. S15D showed the lineage from published *in vivo* meiosis data, while fig. S16 comes from our *in vitro* meiosis data.

16. It has not been mentioned how the authors address the drop-outs in single-cell RNA-seq data which is an important issue in single-cell RNA-seq data analysis.

Stringent gene selection criteria and dimensional reduction were used through the Seurat package, which substantially mitigates the issues implicated by drop-outs. Although we do not use a specific method to impute drop-outs, we focus on highly variable genes, which are less effected by drop-outs.

17. The authors provided a GitHub repository (<https://github.com/iamzhangxiaoyu/scRNA-seq>;) for the source of codes, however, no codes can be seen there (July 25, 2020, 9:00AM).

We have uploaded the source of codes and updated the GitHub.

18. The authors need to add substantial citations for all the packages and methods they applied and specify the versions of each of the packages. For example, Seurat, JackStraw procedure, tSNE, Monocle, Slingshot, Destiny, pheatmap, etc..

Relevant citations have now been added to our revised manuscript.

Data Availability

1. Authors mentioned “FastQ files of single-cell RNA-seq and bulk RNA-seq are available on GEO GSE” (Page 29, last line) but there is no GEO number provided. Is there a specific reason for that?

The access number is GSE153300, and the reviewer password is: sradsombhqhdkp.

REVIEWER COMMENTS

Reviewer #1 (Remarks to the Author):

Overall, the authors made a number of revisions to improve the manuscript, which remains quite interesting. However there are issues that still remain (relevant to my original major points), though I don't think they are enormously serious. The numbering scheme below corresponds to the original major points, and I provide my opinion on how the authors addressed them.

1) The nature of "nutrient restriction" (NS) is still unclear. My question of what nutrient is being restricted was never addressed? At least they could speculate.

2) OK.

3) OK.

4) OK.

5) OK.

6) OK (curious about what melatonin is doing).

7) OK.

8) Unsatisfactory response. There are several datasets listing recombination hotspots and SPO11 cleavage sites in mice. The comparisons should be simple, and could lend much strength to the paper.

9) OK

10) It is hard to believe that the authors did not try culturing longer to see if the cells could progress through meiotic divisions. There really should be a comment on this even if data is not presented.

11) OK

12) OK

Reviewer #2 (Remarks to the Author):

overall sufficed my previous questions and concerns. I suppose that the authors indeed observed the meiotic initiation of SSC in vitro in a certain culture condition. However, given that the culture conditions were now disclosed, there still remain concerns to be addressed and corrected in their interpretation of the data. According to their revised Table S8, the meiotic progression medium was formulated by omitting GDNF and FGF2 from the complete SSC medium. Because they did not properly disclose it in the previous manuscript, now this description raises another possibility that the signaling under the presence or absence of GDNF and FGF2 rather than "the nutrient restriction" may have more direct effects on the cell fate direction to mitotic growing of SSC or to differentiation into spermatocyte. So, it is not clear if nutrient restriction was simply responsible for meiotic initiation. Because this is the crucial point for interpreting their study, this should be unambiguously clarified. As the authors state, "the nutrient restriction" condition activates autophagy but not for meiotic initiation. Moreover, as the authors showed, not only "the nutrient restriction" but also RA is required for meiotic initiation. Thus, the correct interpretation that can be drawn from their data is that "the nutrient restriction" is not sufficient for meiotic initiation. Further, it is known that the synthetic nitrogen starved condition triggers spore formation program including meiosis in budding and fission yeast. Thus, "inducer of yeast meiosis" does not equal to what the authors refer to "the nutrient restriction" condition. Thus, the title is still misleading. This should be corrected.

Minor comment.

legends for Fig S9 C D E F are missing.

Reviewer #3 (Remarks to the Author):

The manuscript has been improved after the revision. I appreciate the authors' effort. Some of the comments have been addressed, but for some comments, the authors replied that they made revisions accordingly but the revisions are not shown in the revised manuscript. The following comments should be addressed before it meets publication requirements. The numbering is the same as in previous comments, so the authors could refer to the response letter for more details.

Analysis of single-cell RNA-seq data

5. The authors replied, "we used a global-scaling normalization method "LogNormalize" that normalizes the feature expression measurements for each cell by the total expression, multiplies this by a scale factor (10,000 by default), and log-transforms the result. " This normalization step should be specified in the Methods section, but please rephrase this sentence since it is the same as the Seurat online tutorial (https://satijalab.org/seurat/v3.2/pbmc3k_tutorial.html). Authors could replace the sentence "Seurat object was built after filtered and normalized with default settings. " with the normalization steps. Seurat objects have been created before the authors can apply `Seurat::NormalizeData()` for normalization. Besides, after the authors have added the data normalization steps in Methods, it is recommended to use "expression" instead of "relative expression" to refer to gene expression, since 1) this normalization step is quite commonly-used and 2) it can avoid confusions when seeing "relative levels" somewhere else (e.g. S11).

9. The authors replied, "We have updated the word "feature" to "number of genes" in our revised manuscript", but actually it is not. In section Analysis of single-cell RNA-seq data, line 4, it is still there.

10. The authors replied, "In the beginning, we used the default 2,000 highly variable genes for downstream analysis in Seurat. However, we found there was no effect on clusters compared with using 6,000 highly variable genes. Thus, to preserve more interesting biological structures, we used 6,000, instead of 2,000". However, it is still not convincing to call all these genes as highly-variable since many of them have variance around 0 (Fig. S9E). Authors could consider rephrase it.

12. The authors replied that the sentence "Cluster define was by specific marker genes using "FeaturePlot" function" (Page 6, section Analysis of single-cell RNA-seq data, line 8) has been corrected as "Clusters were defined by specific markers by using 'FeaturePlot' function", but actually it is not. The original sentence is still there.

14. Authors replied, "As differential gene expression analysis is built by using the Monocle in our study, we have transferred the cluster identified in Seurat to cell types in Monocle. The code used is shown in github. ". It is unclear where in the github should we go to find out the codes. Does it mean that the authors performed differential gene expression analysis and manually annotated the cell types for the clusters using marker genes? If yes, for the convenience of readers, authors could consider clarifying these using texts in the Methods section instead of referring readers to codes. The original sentence "Differential gene expression analysis for different cell types was by using the Monocle" (grammatically incorrect) should be revised. More details should also be included on how the differential test is performed.

15. The authors replied, "Different datasets are presented in figs. S15 and S16: fig. S15D showed the lineage from published in vivo meiosis data, while fig. S16 comes from our in vitro meiosis data. ", but it is unclear how the clustering analysis was performed. The authors wrote, "We merged the scRNA-seq data obtained from our in vitro meiosis study with a published data of in vivo meiosis during spermatogenesis. Then we calculate the mean expression levels of genes in each cluster using monocle2 as show in github. We used the 'ComBat' function from the R/Bioconductor sva package to regress-out the technical effect." but it is unclear 1) what is "merged" referring to? Does it mean that

two single-cell RNA-seq datasets are integrated (e.g., using Seurat, or Harmony)? If yes, why there is a "combat" step? If no, how did the authors address the technical difference due to study difference? 2) does any evidence or previous reference exist to show that it is valid to perform monocle2 for differential analysis in pseudobulks other than in single cells?

16. The authors replied, "Stringent gene selection criteria and dimensional reduction were used through the Seurat package, which substantially mitigates the issues implicated by drop-outs.". However, it was only mentioned that authors applied default Seurat filtering steps (Genes expressed in less than 3 cells were filtered out and cells with expressed genes less than 200 were excluded) which is quite different from addressing drop-out events in scRNA-seq data. The authors are recommended to plot (e.g., a violin plot, boxplot, or others) the distribution of zero-expression (the percentage cells where a gene has no expression) to justify this? Besides, many imputation methods have been developed to address drop-out events in scRNA-seq data (<https://doi.org/10.1186/s13059-020-1926-6>); alternatively, authors could leverage the conclusion from a recent benchmark of scRNA-seq methods (<https://doi.org/10.1186/s13059-020-02132-x>) that "the majority of the methods did not improve performance in downstream analyses compared to no imputation, in particular for clustering and trajectory analysis, and thus should be used with caution." Either way should be specified in the manuscript.

18. The authors replied, "Relevant citations have now been added to our revised manuscript.", but many packages are still not cited. For example, JackStraw procedure, tSNE, pheatmap, clusterProfiler, etc. It is also recommended to include the versions of the packages.

Data Availability

The authors have addressed my comments by sharing the GSE number for the fastq files.

Point-to-point response to reviewers' comments

We thank all reviewers for the time and effort in reviewing our revised manuscript.

Reviewer #1

Overall, the authors made a number of revisions to improve the manuscript, which remains quite interesting. However, there are issues that still remain (relevant to my original major points), though I don't think they are enormously serious. The numbering scheme below corresponds to the original major points, and I provide my opinion on how the authors addressed them:

1) The nature of "nutrient restriction" (NS) is still unclear. My question of what nutrient is being restricted was never addressed? At least they could speculate.

We speculate that amino acid restriction could be a nutrient signal for meiotic initiation in the mammalian system. The following text has been added to the Discussion section of our manuscript (line 393 – 402):

"The nature of the nutrient restriction condition – that is, restriction of which nutrient type(s) is necessary for meiotic induction in the mammalian system – remains to be further defined. In yeasts, nitrogen starvation triggers meiosis. It appears that some yeast factors that respond to nitrogen starvation have become sensitive to amino acid starvation in mammalian cells. For instance, Dhh1 responds to nitrogen starvation in yeasts, while its mammalian homolog, Ddx6, responds to amino acid starvation (ref. 50). Interestingly, past studies that characterized the fluid composition inside seminiferous tubules where meiosis takes place have showed that it is low in most amino acids compared to blood plasma (ref. 51). Consistently, EBSS used in our nutrient restriction medium applies robust amino acid starvation to cells. Thus, we speculate that amino acid restriction is a critical nutrient signal for mammalian meiotic initiation."

2) OK.

3) OK.

4) OK.

5) OK.

6) OK (curious about what melatonin is doing).

As an antioxidant and anti-inflammatory molecule, melatonin was included that may protect male germ cells under stress conditions (Zhang *et al*, Free Radic Biol Med 2019 **137**: 74).

7) OK.

8) Unsatisfactory response. There are several datasets listing recombination hotspots and SPO11 cleavage sites in mice. The comparisons should be simple and could lend much strength to the paper.

Thank you for the suggestion. A recent study has shown that the bindings of SPO11, DMC1, RAD51, and RPA2 are highly overlapped at the recombination hotspots (Hinch *et al*, Mol Cell 2020 **79**: 689). In our manuscript, to examine the genomic distribution of meiotic DSBs formed *in vitro*, we selected the strong recombination hotspots from a published DMC1 ChIP-seq database (mm9 as reference genome; Smagulova *et al*, Nature 2011 **472**: 375) and used DMC1 ChIP. As the reviewer suggested, to confirm that these strong recombination hotspots are SPO11 cleavage sites, we converted the SPO11-oligo dataset (Lange *et al*, Cell 2016 **167**: 695), which was mapped to mm10 as reference genome, to mm9 as reference genome. We show that SPO11 binds near to these strong recombination hotspots (new fig. S22B). This data lends further support to our data that NRRRA-induced meiotic DSBs *in vitro* are cleaved by SPO11 (**Fig. 1D**). This information has been added to the revised manuscript (lines 329 – 331).

9) OK

10) It is hard to believe that the authors did not try culturing longer to see if the cells could progress through meiotic divisions. There really should be a comment on this even if data is not presented.

We indeed haven't tried culturing longer to see whether cells progress through meiotic divisions. We have devoted all our efforts to examine meiotic initiation *in vitro*, which has allowed us to characterize the transcriptomic and genomic events during this process in greater details. These include the use of both bulk RNA-seq and single cell RNA-seq, the generation of *Stra8*-deficient, *Spo11*-deficient, and RFP-GFP-LC3 autophagy reporter primary SSC cultures, and the characterization of meiotic DSB formed *in vitro*. We will examine whether cells could progress through meiotic division in our follow-up study.

11) OK

12) OK

Reviewer #2

Overall sufficed my previous questions and concerns. I suppose that the authors indeed observed the meiotic initiation of SSC *in vitro* in a certain culture condition. However, given that the culture conditions were now disclosed, there still remain concerns to be addressed and corrected in their interpretation of the data. According to their revised Table S8, the meiotic progression medium was formulated by omitting GDNF and FGF2 from the complete SSC medium. Because they did not properly disclose it in the previous manuscript, now this description raises another possibility that the signaling under the presence or absence of GDNF

and FGF2 rather than “the nutrient restriction” may have more direct effects on the cell fate direction to mitotic growing of SSC or to differentiation into spermatocyte. So, it is not clear if nutrient restriction was simply responsible for meiotic initiation. Because this is the crucial point for interpreting their study, this should be unambiguously clarified.

The reviewer was correct that “the meiotic progression medium was formulated by omitting GDNF and FGF2 from the complete SSC medium”. To clarify that GDNF and FGF2 do not regulate meiosis directly, we revised lines 260 – 261 and lines 263 – 265 in the manuscript.

We’d like to kindly emphasize that, as shown by the experimental workflow in Figs. 2A, 3A and 3D, this “meiotic progression” medium was used after nutrient restriction with RA (NRRRA)-induced meiotic initiation to support meiotic progression. In Fig. 1, we show that NRRRA induces meiotic initiation; neither nutrient restriction nor RA alone exerts this effect. In Fig. 3D, we show that this “meiotic progression” medium had no direct effect on meiosis in cultured SSCs that had not received NRRRA treatment to induce meiotic initiation – that is, if cultured SSCs had not received NRRRA-induced meiotic initiation, treatment of this “meiotic progression” medium did not cause meiosis. Moreover, in our experiment using F9 cells that do not require GDNF and FGF2 in the culture system, NRRRA was sufficient to induce meiotic gene activation (fig. S2), suggesting that this process does not involve GDNF and FGF2. Furthermore, past literatures have shown that the targets of GDNF and FGF2 are mostly related to cell cycle regulators and SSC self-renewal, but not directly related to meiosis (Hofmann *et al*, 2005 Dev Biol **279**: 114; Wu *et al*, Biol Repro 2011 **85**: 1114; Shii *et al*, Development 2012 **139**: 1734; Zhang *et al*, Cell Research 2012 **22**: 773).

As the authors state, “the nutrient restriction” condition activates autophagy but not for meiotic initiation. Moreover, as the authors showed, not only “the nutrient restriction” but also RA is required for meiotic initiation. Thus, the correct interpretation that can be drawn from their data is that “the nutrient restriction” is not sufficient for meiotic initiation. Further, it is known that the synthetic nitrogen starved condition triggers spore formation program including meiosis in budding and fission yeast. Thus, “inducer of yeast meiosis” does not equal to what the authors refer to “the nutrient restriction” condition. Thus, the title is still misleading. This should be corrected.

We have revised the title into: “Nutrient restriction synergizes with retinoic acid to induce mammalian meiotic initiation *in vitro*”.

Minor comment: legends for fig. S9C, D, E, and F are missing.

Legends for fig. S9C, D, E, and F have been added to the revised Supplementary Materials.

Reviewer #3:

The manuscript has been improved after the revision. I appreciate the authors’ effort. Some of the comments have been addressed, but for some comments, the authors replied that they

made revisions accordingly, but the revisions are not shown in the revised manuscript. The following comments should be addressed before it meets publication requirements. The numbering is the same as in previous comments, so the authors could refer to the response letter for more details.

Analysis of single-cell RNA-seq data

5. The authors replied, “we used a global-scaling normalization method "LogNormalize" that normalizes the feature expression measurements for each cell by the total expression, multiplies this by a scale factor (10,000 by default), and log-transforms the result.” This normalization step should be specified in the Methods section, but please rephrase this sentence since it is the same as the Seurat online tutorial (https://satijalab.org/seurat/v3.2/pbmc3k_tutorial.html). Authors could replace the sentence “Seurat object was built after filtered and normalized with default settings.” with the normalization steps. Seurat objects have been created before the authors can apply `Seurat::NormalizeData()` for normalization.

We have revised this section in Methods as below (lines 188 – 194 in the Supplementary Materials):

“Cell clustering was performed by Seurat (<https://satijalab.org/seurat/>, R package, v3.1) (ref. 6). Seurat object was created first. Then, we discarded low-quality cells, in which less than 200 genes were detected. Genes expressed in less than 3 cells were filtered out. We also filtered cells that have lower than 2,000 genes and that contain mitochondrial genome higher than 10% of mapped reads. We then used LogNormalize, a global-scaling normalization method, which normalized gene expression measurements by the total expression per cell, followed by multiplication of the result by a default scale factor (10,000) and subsequent log-transformation...”

Besides, after the authors have added the data normalization steps in Methods, it is recommended to use “expression” instead of “relative expression” to refer to gene expression, since 1) this normalization step is quite commonly-used and 2) it can avoid confusions when seeing “relative levels” somewhere else (e.g. S11).

We have corrected “relative expression” to “gene expression”. Please see line 195 in the Supplementary Materials.

9. The authors replied, “We have updated the word “ufeature” to “number of genes” in our revised manuscript”, but actually it is not. In section Analysis of single-cell RNA-seq data, line 4, it is still there.

The word “2,000 ufeature” has been updated to “2,000 genes”. Please refer to lines 190 – 191 in the Methods section of the Supplementary Materials:

10. The authors replied, “In the beginning, we used the default 2,000 highly variable genes for downstream analysis in Seurat. However, we found there was no effect on clusters compared with using 6,000 highly variable genes. Thus, to preserve more interesting biological structures, we used 6,000, instead of 2,000”. However, it is still not convincing to call all these genes as highly-variable since many of them have variance around 0 (Fig. S9E). Authors could consider rephrase it.

We have rephrased “6,000 highly variable genes” to “top 6,000 variable genes”. Please refer to line 196 in the Methods section of the Supplementary Materials.

12. The authors replied that the sentence “Cluster define was by specific marker genes using “FeaturePlot” function” (Page 6, section Analysis of single-cell RNA-seq data, line 8) has been corrected as “Clusters were defined by specific markers by using ‘FeaturePlot’ function”, but actually it is not. The original sentence is still there.

This sentence has now been updated to “Cluster definition was performed by specific marker genes and the ‘FindClusters()’ function in Seurat”. Please refer to line 199 – 200 in the Methods section of the Supplementary Materials.

14. Authors replied, “As differential gene expression analysis is built by using the Monocle in our study, we have transferred the cluster identified in Seurat to cell types in Monocle. The code used is shown in github.”. It is unclear where in the github should we go to find out the codes. Does it mean that the authors performed differential gene expression analysis and manually annotated the cell types for the clusters using marker genes? If yes, for the convenience of readers, authors could consider clarifying these using texts in the Methods section instead of referring readers to codes. The original sentence “Differential gene expression analysis for different cell types was by using the Monocle” (grammatically incorrect) should be revised. More details should also be included on how the differential test is performed.

The reviewer is correct that we manually annotated cells in Clusters 0 – 3 as germ cells based on germ cell marker gene, *Ddx4*, expression and cells in Cluster 4 as mouse embryonic fibroblast (MEF) based on fibroblast marker gene, *S100a4*, expression. This information is noted in lines 182 – 184 of our manuscript and has been added to lines 199 – 202 in the Methods section of the Supplementary Materials.

We have corrected “differential gene expression analysis” to “gene expression analysis”. Thus, the corresponding sentence has been updated to “Gene expression analysis for different clusters were performed by using the Monocle...”. Please refer to lines 202 – 203 in the Methods section of the Supplementary Materials.

For the gene expression analysis, we used a code from a published study (Stévant *et al.* Cell Reports 2018 **22**: 1589) with modifications. This code (shown below) is available at: <https://github.com/iamzhangxiaoyu/scRNA-seq/blob/master/DEG.R>.

```

library(Seurat)
library(monocle)
load(file = 'aggr_filter_marker.output.Rdata')
tmp=read.csv('clustering.csv')
clustering=tmp[,2];names(clustering)=tmp[,1]
table(clustering)
count_matrix <- aggr@assays$RNA@counts
stages <- aggr@meta.data$TimePoints
table(stages)
We used clustering, count_matrix and stages as input to monocle2.
expr_matrix <- as.matrix(count_matrix)
sample_sheet <- data.frame(cells=colnames(count_matrix),
                           stages=stages,
                           cellType=clustering)
rownames(sample_sheet)<- colnames(count_matrix)
gene_annotation <- as.data.frame(rownames(count_matrix))
rownames(gene_annotation)<- rownames(count_matrix)
colnames(gene_annotation)<- "genes"
pd <- new("AnnotatedDataFrame", data = sample_sheet)
fd <- new("AnnotatedDataFrame", data = gene_annotation)

HSMM <- newCellDataSet(
  as(expr_matrix, "sparseMatrix"),
  phenoData = pd,
  featureData = fd,
  lowerDetectionLimit=0.5,
  expressionFamily=negbinomial.size()
)

HSMM <- detectGenes(HSMM, min_expr = 5)
# HSMM <- HSMM[fData(HSMM)$num_cells_expressed > 5, ]
HSMM <- HSMM[fData(HSMM)$num_cells_expressed > 10, ]

HSMM <- estimateSizeFactors(HSMM)
HSMM <- estimateDispersions(HSMM)

diff_test_res <- differentialGeneTest(
  HSMM,
  fullModelFormulaStr="~cellType",
  cores = 4
)

sig_genes_0.05 <- subset(diff_test_res, qval < 0.05)
sig_genes_0.01 <- subset(diff_test_res, qval < 0.01)

print(paste(nrow(sig_genes_0.05), " significantly DE genes (FDR<0.05).",
sep=""))
print(paste(nrow(sig_genes_0.01), " significantly DE genes (FDR<0.01).",
sep=""))

diff_test_res <- subset(diff_test_res, qval< 0.01)

save(diff_test_res,file='diff_test_res.Rdata')
save(HSMM, file = 'HSMM.Rdata')
load('HSMM.Rdata')

```

```

load('diff_test_res.Rdata')

cluster_nb <- unique(clustering)
mean_per_cluster <- vector()
diff_test_res <- diff_test_res[order(rownames(diff_test_res)),]
count_matrix <- count_matrix[order(rownames(count_matrix)),]
count_de_genes <- count_matrix[rownames(count_matrix) %in%
diff_test_res$genes,]
print(dim(count_de_genes))
for (clusters in cluster_nb) {
  # print(head(count_de_genes[,
  #           colnames(count_de_genes) %in%
names(clustering[clustering==clusters])
  #   ]))
  mean <- rowMeans(
    as.matrix(count_de_genes[,
                                colnames(count_de_genes) %in%
names(clustering[clustering==clusters])
                                ]))
  )
  names(mean) <- clusters
  mean_per_cluster <- cbind(
    mean_per_cluster,
    mean
  )
}
colnames(mean_per_cluster) <- cluster_nb
up_reg_cluster <-
colnames(mean_per_cluster)[apply(mean_per_cluster,1,which.max)]
de_genes_table <- data.frame(
  diff_test_res,
  mean_per_cluster,
  cluster=up_reg_cluster
)
write.csv(de_genes_table, quote = FALSE, file=
"DE_genes_per_clusters_4_groups.csv")

```

15. The authors replied, “Different datasets are presented in figs. S15 and S16: fig. S15D showed the lineage from published *in vivo* meiosis data, while fig. S16 comes from our own *in vitro* meiosis data”, but it is unclear how the clustering analysis was performed. The authors wrote, “We merged the scRNA-seq data obtained from our *in vitro* meiosis study with a published data of *in vivo* meiosis during spermatogenesis. Then we calculate the mean expression levels of genes in each cluster using monocle2 as show in github. We used the ‘ComBat’ function from the R/Bioconductor sva package to regress-out the technical effect.” but it is unclear: 1) what is “merged” referring to? Does it mean that two single-cell RNA-seq datasets are integrated (e.g., using Seurat, or Harmony)? If yes, why there is a “combat” step? If no, how did the authors address the technical difference due to study difference? 2) does any evidence or previous reference exist to show that it is valid to perform monocle2 for differential analysis in pseudobulks other than in single cells?

To perform clustering analysis, we used two datasets: 1) our own scRNA-seq dataset on *in vitro* meiosis reported in this manuscript, which detected 18,088 cells, and 2) a published dataset on

in vivo mouse meiosis (Hermann *et al.* Cell Reports, 2018 **25**:1650; ref. 15 in Methods). Of note, because a scRNA-seq dataset that is focused on early *in vivo* meiosis is not available, we used this published dataset that contains cells during entire spermatogenesis. Thus, majority of the cells is post-meiotic spermatids, and only 72 premeiotic and meiotic cells are relevant to our study. This difference in cell numbers from two datasets imposes a challenge to conduct clustering analysis using Seurat or Harmony.

To solve this problem, we modified a method published in a recent paper (Nature Cell Biology, 2019 **21**: 835; now included as ref. 16 in Methods). This method used the 'ComBat' function from the R/Bioconductor sva package to regress-out the technical effect and compare the mean expression profile of the clusters identified by scRNA-seq to the expression profile of bulk RNA-seq.

Thus, we calculated the mean expressions of genes in each cluster from the *in vitro* and the *in vivo* meiosis datasets as “pseudo-bulks”, and then used the same protocol to remove technical effect and conduct PCA analysis (calculated using the 'prcomp' R function) as “pseudo-bulks” rather than as single cells.

fig. S15 and fig. S16 represent two different datasets: fig. S15 shows our analysis of the premeiotic and meiotic cells selected from the *in vivo* dataset (Hermann *et al.* Cell Reports, 2018 **25**: 1650; ref. 15 in Methods), and fig. S16 shows the pseudotime analysis of our *in vitro* meiosis dataset.

16. The authors replied, “Stringent gene selection criteria and dimensional reduction were used through the Seurat package, which substantially mitigates the issues implicated by drop-outs.”. However, it was only mentioned that authors applied default Seurat filtering steps (Genes expressed in less than 3 cells were filtered out and cells with expressed genes less than 200 were excluded) which is quite different from addressing drop-out events in scRNA-seq data. The authors are recommended to plot (e.g., a violin plot, boxplot, or others) the distribution of zero-expression (the percentage cells where a gene has no expression) to justify this?

Besides, many imputation methods have been developed to address drop-out events in scRNA-seq data (<https://doi.org/10.1186/s13059-020-1926-6>); alternatively, authors could leverage the conclusion from a recent benchmark of scRNA-seq methods (<https://doi.org/10.1186/s13059-020-02132-x>) that “the majority of the methods did not improve performance in downstream analyses compared to no imputation, in particular for clustering and trajectory analysis, and thus should be used with caution.” Either way should be specified in the manuscript.

We have considered potential issues caused by drop-outs during scRNA-seq. Our scRNA-seq analysis is focused on clustering and trajectory analyses. As the reviewer kindly pointed out, most methods to compensate for drop-outs by imputation do not improve performance, particularly in clustering and trajectory analyses (Hou *et al.* Genome Biol 2020 **21**: 218). Moreover, most published scRNA-seq studies on clustering and trajectory analyses also do not

use imputation to address drop-outs. Thus, we applied stringent gene selection criteria and dimensional reduction during filtration process as described in lines 189 – 199.

In addition, we have plotted hist plots for drop-out rates of marker genes, which show that the drop-out rates are low.

18. The authors replied, “Relevant citations have now been added to our revised manuscript.”, but many packages are still not cited. For example, JackStraw procedure, tSNE, pheatmap, clusterProfiler, etc. It is also recommended to include the versions of the packages.

We have double-checked the manuscript to ensure all relevant citations, including available version information, added to our manuscript. Please refer to references 3 (DESeq2), 4 (clusterProfiler), 5 (GSVA), 6 (Seurat; version 3.1), 7 (PCA), 8 (JackStraw), 9 (t-SNE), 10 (Monocle2, version 2.12), 12 (Destiny), 13 (Slingshot), 14 (pheatmap), and 17 (sva).

REVIEWER COMMENTS

Reviewer #1 (Remarks to the Author):

The authors have addressed my remaining concerns adequately. They also seemed to do a reasonable job with Reviewer#2's concerns, though I am not sufficiently versed in single cell data analysis to comment on the revisions made in response to Rev #3.

Reviewer #2 (Remarks to the Author):

All of my concerns have been addressed in the revisions.

Reviewer #3 (Remarks to the Author):

I appreciate the authors' effort in revising the writing of the methods section. Many of the previously unclear sentences have been addressed. By following the analysis steps that the authors have written, some results seem a bit fishier.

The authors wrote, "We then used LogNormalize, a global-scaling normalization method, which normalized gene expression measurements by the total expression per cell, followed by multiplication of the result by a default scale factor (10,000) and subsequent log-transformation". In Figure 2F, the expression level of Sycp3 can reach higher than 100, meaning the original read count for that gene is 2^{100} , i.e. 10^{30} (if base 2 was used in log; or 10^{100}). Similarly, in Figure S14B, the expression of those genes are shown to be 30 to 100. If that is true, the cells library size is large than 10^{30} which is not a typical number for single cells sequenced using 10x platform. Is it possible that there are some droplet, triplet, or an inappropriate scale factor was used, or the some of the processing steps were missed? If so, it could potentially affect many of the analysis results, and could also be part of the reason why "In the beginning, we used the default 2,000 highly variable genes for downstream analysis in Seurat. However, we found there was no effect on clusters compared with using 6,000 highly variable genes."

The authors replied, "we calculated the mean expressions of genes in each cluster from the in vitro and the in vivo meiosis datasets as "pseudo-bulks", and then used the same protocol to remove technical effect and conduct PCA analysis (calculated using the 'prcomp' R function) as "pseudo-bulks" rather than as single cells.". It is (1) unclear whether they normalize the cluster-mean with the single cells in the published data ((Hermann et al. Cell Reports, 2018 25:1650), or they also clustered the published data; (2) not consistent with the main text which is "we performed principal component analysis (PCA) analysis with a published scRNA-seq database (fig. S15) (ref. 31) " (main line 212).

The authors revised the methods section, but the title "Clustering of scRNA-seq data" was not consistent with the paragraph below it "We merged the scRNA-seq data obtained from our in vitro meiosis study with a published data of in vivo meiosis during spermatogenesis. To do this, we modified a method from a published study, in which we calculate the mean expression levels of genes in each cluster using Monocle2 as show in github. We used the 'ComBat' function from the R/Bioconductor sva package to regress-out the technical effect. The first two PCs of the corrected expression were calculated using the 'prcomp' R function. "

Point-by-point response to Reviewer 3' comments

I appreciate the authors' effort in revising the writing of the methods section. Many of the previously unclear sentences have been addressed. By following the analysis steps that the authors have written, some results seem a bit fishier.

We sincerely thank Reviewer #3 for the time and efforts in reviewing our manuscript.

1. The authors wrote, "We then used LogNormalize, a global-scaling normalization method, which normalized gene expression measurements by the total expression per cell, followed by multiplication of the result by a default scale factor (10,000) and subsequent log-transformation". In Figure 2F, the expression level of Sycp3 can reach higher than 100, meaning the original read count for that gene is 2^{100} , i.e. 10^{30} (if base 2 was used in log; or 10^{100}). Similarly, in Figure S14B, the expression of those genes are shown to be 30 to 100. If that is true, the cells library size is large than 10^{30} which is not a typical number for single cells sequenced using 10x platform. Is it possible that there are some droplet, triplet, or an inappropriate scale factor was used, or the some of the processing steps were missed? If so, it could potentially affect many of the analysis results, and could also be part of the reason why "In the beginning, we used the default 2,000 highly variable genes for downstream analysis in Seurat. However, we found there was no effect on clusters compared with using 6,000 highly variable genes."

Thank you for the comment. We would like to kindly remind the reviewer that the violin plots presented in our manuscript were generated by using parameter, slot = "counts" (see below). Thus, the scale factors in Y-axis of the violin plots represent original read counts, i.e., "100" indicates original read counts of 100.

By using Sycp3 as an example, the code for violin plot (available in github) is:

```
library(Seurat)
## Warning: package 'Seurat' was built under R version 3.6.2
library(ggplot2)
## Warning: package 'ggplot2' was built under R version 3.6.2
load(file = 'aggr_filter_marker.output.Rdata')
aggr$seurat_clusters <- factor(aggr$seurat_clusters, levels = c('0', '2', '1', '3', '4'))
aggr@active.ident <- factor(aggr@active.ident, levels = c('0', '2', '1', '3', '4'))
new.cluster.ids <- c("Cluster0", "Cluster1", "Cluster2", "Cluster3", "Cluster4")
names(new.cluster.ids) <- levels(aggr)
aggr <- RenameIdents(aggr, new.cluster.ids)
colP <- c('#560047', '#a53bad', '#eb6bac', '#ffa8a0', '#f7dad7')
load(file = 'Testis2output.Rdata')
VlnPlot(aggr, features = "Sycp3", slot = "counts", log = TRUE, cols = colP, pt.size = 0)
```

The violin plot for Sycp3 generated by using this code is shown below (as in our Fig. 2F).

To check whether these original read counts for meiotic gene expression *in vitro* are within a reasonable range, we examined *Sycp3* expression level in a published scRNA-seq database of *in vivo* meiosis (Hermann *et al.* Cell Reports, 2018 25:1650).

```
VlnPlot(Testis, features = "Sycp3", slot = "counts", log = TRUE, pt.size = 0)
```

The median values of the original read counts of *Sycp3* are comparable between our *in vitro* study and the published *in vivo* study.

2. The authors replied, “we calculated the mean expressions of genes in each cluster from the *in vitro* and the *in vivo* meiosis datasets as “pseudo-bulks”, and then used the same protocol to remove technical effect and conduct PCA analysis (calculated using the 'prcomp' R function) as “pseudo-bulks” rather than as single cells.” It is: (1) unclear whether they normalize the cluster-mean with the single cells in the published data (Hermann *et al.* Cell Reports, 2018 25:1650), or they also clustered the published data; (2) not consistent with the main text which is “we performed principal component analysis (PCA) analysis with a published scRNA-seq database

(fig. S15) (ref. 31)” (main line 212).

We appreciate the reviewer’s comment.

(1) the reviewer is correct that we calculated the cluster-mean by using Monocle2 for each cluster in the published scRNA-seq dataset on in vivo meiosis. The same code was also used to calculate the cluster-mean for each cluster in the dataset of in vitro meiosis. The code is presented below (available in github):

```
library(Seurat)
library(monocle)
load(file = 'aggr_filter_marker.output.Rdata')
tmp=read.csv('clustering.csv')
clustering=tmp[,2];names(clustering)=tmp[,1]
table(clustering)
count_matrix <- aggr@assays$RNA@counts
stages <- aggr@meta.data$TimePoints
table(stages)

expr_matrix <- as.matrix(count_matrix)
sample_sheet <- data.frame(cells=colnames(count_matrix),
                           stages=stages,
                           cellType=clustering)
rownames(sample_sheet)<- colnames(count_matrix)
gene_annotation <- as.data.frame(rownames(count_matrix))
rownames(gene_annotation)<- rownames(count_matrix)
colnames(gene_annotation)<- "genes"
pd <- new("AnnotatedDataFrame", data = sample_sheet)
fd <- new("AnnotatedDataFrame", data = gene_annotation)

HSMM <- newCellDataSet(
  as(expr_matrix, "sparseMatrix"),
  phenoData = pd,
  featureData = fd,
  lowerDetectionLimit=0.5,
  expressionFamily=negbinomial.size()
)

HSMM <- detectGenes(HSMM, min_expr = 5)
# HSMM <- HSMM[fData(HSMM)$num_cells_expressed > 5, ]
HSMM <- HSMM[fData(HSMM)$num_cells_expressed > 10, ]

HSMM <- estimateSizeFactors(HSMM)
HSMM <- estimatedDispersions(HSMM)

diff_test_res <- differentialGeneTest(
  HSMM,
  fullModelFormulaStr="~cellType",
  cores = 4
)

sig_genes_0.05 <- subset(diff_test_res, qval < 0.05)
sig_genes_0.01 <- subset(diff_test_res, qval < 0.01)

print(paste(nrow(sig_genes_0.05), " significantly DE genes (FDR<0.05).", sep=""))
print(paste(nrow(sig_genes_0.01), " significantly DE genes (FDR<0.01).", sep=""))

diff_test_res <- subset(diff_test_res, qval< 0.01)

save(diff_test_res,file='diff_test_res.Rdata')
save(HSMM, file = 'HSMM.Rdata')
load('HSMM.Rdata')
load('diff_test_res.Rdata')
```

```

cluster_nb <- unique(clustering)
mean_per_cluster <- vector()
diff_test_res <- diff_test_res[order(rownames(diff_test_res)),]
count_matrix <- count_matrix[order(rownames(count_matrix)),]
count_de_genes <- count_matrix[rownames(count_matrix) %in% diff_test_res$genes,]
print(dim(count_de_genes))
for (clusters in cluster_nb) {
  # print(head(count_de_genes[,
  #           colnames(count_de_genes) %in% names(clustering[clustering==clusters])
  #           ]))
  mean <- rowMeans(
    as.matrix(count_de_genes[,
                        colnames(count_de_genes) %in%
names(clustering[clustering==clusters])
                        ]))
  )
  names(mean) <- clusters
  mean_per_cluster <- cbind(
    mean_per_cluster,
    mean
  )
}
colnames(mean_per_cluster) <- cluster_nb
up_reg_cluster <- colnames(mean_per_cluster)[apply(mean_per_cluster,1,which.max)]
de_genes_table <- data.frame(
  diff_test_res,
  mean_per_cluster,
  cluster=up_reg_cluster
)
write.csv(de_genes_table, quote = FALSE, file= "DE_genes_per_clusters_4_groups.csv")

```

(2) We have revised the main text (line 210 – 215) as the following to increase clarity:

“To assess the relationship between NRRA-induced meiotic initiation and progression in vitro with those during in vivo meiosis, we used a scRNA-seq dataset of mouse spermatogenesis (fig. S15) (ref. 31). Principal component analysis (PCA) analysis shows that the transcriptional profiles of Clusters 0 to 3 associate with the stages of meiotic initiation (leptotene, the first stage of meiotic prophase) and progression (zygotene and early pachytene) during in vivo spermatogenesis (Fig. 2G).”

To conduct PCA analysis, we calculated cluster mean for each cell cluster in both in vitro and in vivo scRNA-seq datasets by using Monocle2 as shown above. Then we gathered all cluster mean values in both in vitro meiosis (4 cell clusters; circles; see graph on the next page) and in vivo meiosis (5 cell clusters; triangles; see graph on the next page). After applying the ‘ComBat’ function from the R/Bioconductor sva package to regress-out the technical effect, we conducted PCA analysis, which produced the values below:

	PC1	PC2	group
Undifferentiated.Spermatogonial	-17.83	-8.26	in vivo meiosis
Differentiating.Preleptotene.Spermatocytes	-13.93	2.30	in vivo meiosis
Leptotene.Zygotene.Spermatocytes	-1.34	2.95	in vivo meiosis
Early.Pachytene.Spermatocytes	9.28	5.24	in vivo meiosis
Late.Pachytene.Spermatocytes	11.16	9.98	in vivo meiosis
Diplotene.Spermatocytes	15.73	10.35	in vivo meiosis

Cluster0	-19.98	-9.05	in vitro meiosis
Cluster1	-10.94	-2.37	in vitro meiosis
Cluster2	4.05	0.43	in vitro meiosis
Cluster3	7.96	8.58	in vitro meiosis

Then, we used ggplot2 to generate the graph for PCA analysis as shown below:

We have also revised the Methods (line 211 – 218) to provide additional details (please refer to comment 3 below).

3. The authors revised the methods section, but the title “Clustering of scRNA-seq data” was not consistent with the paragraph below it “We merged the scRNA-seq data obtained from our *in vitro* meiosis study with a published data of *in vivo* meiosis during spermatogenesis. To do this, we modified a method from a published study, in which we calculate the mean expression levels of genes in each cluster using Monocle2 as show in github. We used the ‘ComBat’ function from the R/Bioconductor sva package to regress-out the technical effect. The first two PCs of the corrected expression were calculated using the ‘prcomp’ R function.”

We have updated the subheading to “**Relationship analysis of cell clusters in *in vitro* meiosis and *in vivo* meiosis**” in the Methods and revised this section to include additional details in the Methods (line 211 – 218 in Methods).

Relationship analysis of cell clusters in *in vitro* meiosis and *in vivo* meiosis

For *in vivo* meiosis, we used a published scRNA-seq on mouse spermatogenesis (ref. 15 in Methods). To analyze the relationship between the cell clusters identified by scRNA-seq during *in vitro* meiosis with those during *in vivo* meiosis, we used a method from a published study with modifications (ref. 16). Specifically, we calculated the mean expressions of genes in each cell cluster as “pseudo-bulks” by using Monocle2. After gathering all cluster mean values in both *in vitro* meiosis (4 cell clusters) and *in vivo* meiosis (5 cell clusters), ‘ComBat’ function from the R/Bioconductor sva package was applied to regress-out the technical effect (ref. 17). Then, the first two PCs of the corrected expression were calculated by using the ‘prcomp’ R function. The plot for PCA analysis was generated by ggplot2 (ref. 18).

REVIEWERS' COMMENTS

Reviewer #3 (Remarks to the Author):

I appreciate the authors' efforts in revising the manuscript. The analysis process now seems appropriate to me. Here are some minor comments.

1. The authors have presented that they normalized and log-transformed the UMI counts and analyses were based on that scale, but in some plots, the counts were presented as "expression" while in others the log-normalized values were presented as "expression". The authors are highly recommended to re-label all plots where they use gene expression levels to avoid any confusion, i.e. label the corresponding axis as "UMI count" if they use counts and use "Expression" or "Log-normalized count"(or another well-defined word) if they use the log-normalized UMI count values.
2. Line 194, "after filtration" - authors might mean "after filtering"?

Reviewer #3 (Remarks to the Author):

I appreciate the authors' efforts in revising the manuscript. The analysis process now seems appropriate to me. Here are some minor comments.

Thank you for reviewing our manuscript.

1. The authors have presented that they normalized and log-transformed the UMI counts and analyses were based on that scale, but in some plots, the counts were presented as "expression" while in others the log-normalized values were presented as "expression". The authors are highly recommended to re-label all plots where they use gene expression levels to avoid any confusion, i.e. label the corresponding axis as "UMI count" if they use counts and use "Expression" or "Log-normalized count" (or another well-defined word) if they use the log-normalized UMI count values.

We have updated our figure legends to indicate that "expression levels are calculated by UMI count".

2. Line 194, "after filtration" - authors might mean "after filtering"?

We have corrected this word.